

**Observation-based implementation of ecophysiological processes for a rubber plant**
**functional type in the community land model (CLM4.5-rubber_v1)**
Ashehad A. Ali[1], Yuanchao Fan[2], Marife D. Corre[3], Martyna M. Kotowska[4], E. Hassler[3],
Fernando E. Moyano[1], Christian Stiegler[1], Alexander Röll[5], Ana Meijide[6], Andre Ringeler[1],
Christoph Leuschner[4], Tania June[7], Suria Tarigan[8], Holger Kreft[9], Dirk Hölscher[5], Chonggang
Xu[10], Charles D. Koven[11], Rosie Fisher[12], Edzo Veldkamp[3], Alexander Knohl[1]
1. University of Göttingen, Bioclimatology, Göttingen, Germany
2. Uni Research Climate, Bjerknes Centre for Climate Research, Bergen, Norway
3. University of Göttingen, Soil Science of Tropical and Subtropical Ecosystems,
Göttingen, Germany
4. University of Göttingen, Department of Plant Ecology and Ecosystems Research,
Göttingen, Germany
5. University of Göttingen, Tropical Silviculture and Forest Ecology, Göttingen, Germany
6. University of Göttingen, Department of Crop Sciences, Division Agronomy, Göttingen,
Germany
7. Department of Geophysics and Meteorology, Bogor Agricultural University, Bogor,
Indonesia
8. Department of Soil and Natural Resources Management, Bogor Agricultural University,
Bogor, Indonesia
9. University of Göttingen, Biodiversity, Macroecology & Biogeography, Göttingen,
Germany
10. Earth & Environmental Sciences Division, Los Alamos National Laboratory, Los
Alamos, NM, USA





11. Climate and Ecosystem Sciences Division, Lawrence Berkeley National Laboratory,
27        Berkeley, CA, USA

12. Climate and Global Dynamics Laboratory, National Center for Atmospheric Research,
29        Boulder, CO, USA

**Running head:** Rubber plant functional type in the community land model_v4.5



**Abstract**
Land-use change has a strong impact on carbon, energy and water fluxes and its effect is
particularly pronounced in tropical regions. Uncertainties exist in the prediction of future land-
use change impacts on these fluxes by land surface models due to scarcity of suitable measured
data for parametrization and poor representation of key biogeochemical processes associated
with tropical vegetation types. Rubber plantations (*Havea brasilliensis*) are a crucial land-use
type across tropical landscapes that has greatly expanded in recent decades. Here, we first
synthesize the relevant data for describing the biogeochemical processes of rubber from our past
measurement campaigns in Jambi province, Indonesia. We then use these data-sets to develop a
rubber plant functional type (PFT) for the Community Land Model (CLM4.5). Field measured
data from small-holder plantations on leaf litterfall, soil respiration, latex harvest, leaf area index,
transpiration, net primary productivity, and above-ground and fine root biomass were used to
develop and calibrate a new PFT-based model (CLM4.5-rubber).
CLM-rubber predictions adequately captured the annual net primary productivity and
above-ground biomass as well as the seasonal dynamics of leaf litterfall, soil respiration, soil
moisture and leaf area index. All of the predicted water fluxes of CLM-rubber were very similar
to a site-specific calibrated soil water model. Including temporal variations in leaf life span
enabled CLM-rubber to better capture the seasonality of leaf litterfall.
Increased sensitivity of stomata to soil water stress and the enhancement of growth and
maintenance respiration of fine roots in response to soil nutrient limitation enabled CLM-rubber
to capture the magnitude of transpiration and leaf area index. Since CLM-rubber predicted
reasonably well the carbon and water use, we think that the current model can be used for larger-
scale simulations within Jambi province because more than 99% of the rubber plantations are
smallholder owned in Jambi province and have low soil fertility.

**Keywords:** plant functional traits, leaf age, productivity, water use, stomatal conductance





## Introduction


Historical records show that Indonesia has had accelerated rates of land-use change from
forest to croplands due to economic development and policy reforms (Gellert, 2005). Within
Indonesia, Jambi province on Sumatra has been a hotspot of land-use change with a relatively
large area of forest converted to rubber plantations over the past two decades (Melati, 2017), in
part due to projected increases in the demand of this commodity (Eleanor et al., 2015). Little is
known about how these land-use changes alter the biogeochemical processes of the carbon and
water cycles (Mann, 2009; Powers et al., 2011; Qui, 2009), which are fundamental for ecosystem
services. Previous studies have shown that land-use changes to rubber plantation decrease above
and below-ground carbon pools (de Blécourt et al., 2013; Ziegler et al., 2009) and affect the soil
nitrogen cycle (Allen et al., 2015; Corre et al., 2006). Thus, quantifying land-atmosphere
interactions of rubber plantations in the context of ongoing land-use and climate change is
essential for understanding local, regional and even global carbon and water balances.
So far, insufficient field data are the main limiting factor of our current understanding of
carbon and water cycling in rubber plantations (Blagodatsky et al., 2016; Carr, 2012). Although
traditional field-based methods are critical for identifying how biogeochemical processes are
affected by land-use changes to rubber plantations (e.g. Allen et al., 2015), they also have
limitations, especially when analyzing interactions between different processes and extrapolating
values to long-term temporal and large spatial scales. In contrast, remote sensing approaches
provide essential information on past land-use changes and surface properties of rubber
plantations (Ranganath et al., 2004; Senf et al., 2013), but they do not completely describe
ecosystem-scale changes, nor the mechanisms behind the changes. Quantitative understanding of
the physiological processes leading to biogeochemical disruption is critical for making future
projections of the environmental implications associated with different land-use change
scenarios, and that is only possible with modelling techniques such as process-based land surface
models used in conjunction with the data sources described.
Numerous land surface models differ in their prediction of land-use change effects on
carbon (Houghton et al., 2012) and water cycles (Boisier et al., 2012; Pitman et al., 2009). Such
uncertainties in land surface models may stem from errors in measurements of meteorological
variables (Rahul et al., 2014), incorrect initial conditions (Hanna et al., 2017), poor
representation of processes (Ali et al., 2016) or errors in parameters (Bonan and Doney, 2018).



Errors in model parameters are considered to be the largest uncertainty in various land surface

models (Bonan and Doney, 2018), including the Community Land Model (CLM). The CLM

version 4.5, used here, represents naturally- and crop-vegetated land units (Oleson et al., 2013)

as patches of plant functional types (PFTs) defined by key ecological functions (Bonan et al.,

2002). The existing parameterization of CLM allows an adequate description of the specific

land-use change effects on annual and perennial crops (Oleson et al., 2013). However, the

biogeochemical cycles of most of the woody tree crops, including rubber are not yet

implemented in CLM (but see Fan et al., 2015).

Rubber (*Havea brasilliensis*) is a commercially important tree species native to the

Amazon rainforest (Wycherley, 1992) but cultivated throughout the tropics. The species is

evergreen in its native range, but drought deciduous in other tropical regions, including

Indonesia (Kotowska et al., 2016), Thailand (Giambelluca et al., 2016) and China (Lin et al.,

2018). The mechanistic basis for the leaf habit of rubber remains poorly understood. In regions

having a marked dry season, the period of defoliation is short and re-foliation occurs before the

commencement of the rainy season, triggered by an increase in day length (Maite et al., 2008). In

contrast, if the dry season is less pronounced, leaf fall occurs more gradually, new leaves develop

more slowly and, although the trees are never completely leafless, latex yields are reduced more

than in situations where complete defoliation occurs.

In this study, we develop a sub-model called "CLM4.5-rubber_v1", within the framework

of CLM4.5, which simulates the productivity, growth, yield, and water and energy cycles of

rubber. To reflect the specific growth characteristic of rubber trees, we modify and develop the

parameters and processes of the existing tropical deciduous forest PFT. The existing drought-

deciduous phenology scheme of the tropical PFT is modified together with the carbon and

nitrogen allocation module, where carbon exports through latex harvest influence both carbon

and nitrogen allocation.

The main objectives of this paper are to (1) implement phenology, carbon and nitrogen

allocation, and yield dynamics for to represent the physiology of rubber plants in CLM-rubber,

(2) use the developed model to test the hypothesis that drought will reduce the latex yield of

rubber plantation, and (3) use CLM-rubber to generate hypotheses that field experimentalists can

investigate in the future. To achieve these objectives, we synthesized the data collected both by

short-term field campaigns and intensive one-year measurements in small-holder rubber





plantations, which are commonly unfertilized, in Jambi, Indonesia and used part of the data for
calibration and the rest for validation of CLM-rubber.

**Methodology**
*Overview*

We made several modifications to the parametrization of the drought deciduous tropical

PFT and implemented phenology, carbon & nitrogen allocations and latex yield processes as we
developed the rubber PFT in CLM using the measured data. We made these changes in a
systematic way as described below and show the results that includes the effect of the overall
change. To save space, we include figures for the model calibration in the main manuscript and
put figures for the model validation in the supplementary section.

*Study Sites*

Our study site is located on mineral soils (Acrisols) located in the lowlands of Jambi

province, Indonesia (2° 0′ 57″ S, 103° 15′ 33″ E, 40 - 100 m above sea level). The studied rubber
plantations were owned by smallholders who did not fertilize 2-5 years prior to and during our
field measurements that started in 2013. A large part of the Jambi province had been converted
to rubber plantations in the past two decades (Margono et al., 2012), and thus this study area was
selected as a hotspot of rubber expansion by our ongoing collaborative research center
(Ecological and Socio-economic Functions of Tropical Lowland Rainforest Transformation
Systems, http://www.uni-goettingen.de/en/310995.html). The mean annual temperature in Jambi
is 26.7 ± 1.0 °C and the mean annual precipitation is 2235 ± 385 mm (Drescher et al., 2016). The
dry season is usually from July to August and the rainy season occurs from October to April.

Measurements were performed in two landscapes within the Jambi province that differed

mainly in soil texture: loam and clay Acrisol soils (Allen et al., 2015). The loam Acrisol soil was
located about 80 km southwest of Jambi City, and hereafter referred to as the Harapan landscape.
The clay Acrisol soil was located about 90 km west of Jambi City, and hereafter referred to as
the Bukit Duabelas landscape. Within each of the two landscapes, four rubber plantations were
chosen and within each plantation a 50 m x 50 m plot was established, totaling eight plots
(Kotowska et al., 2015). On average, the rubber plantations in Bukit Duabelas landscape were
five years younger than those in Harapan landscape.



We collated the following measured data-sets from each of the eight plots: above-ground
biomass, net primary productivity, leaf litterfall, latex yield, fine root biomass; soil moisture, soil
respiration, leaf area index, and transpiration. All of the data were obtained between 2012 and
2014, except leaf area index, which was measured in 2018. Additional information on vegetation
characteristics such as rubber tree density, tree height, and basal area can be found in Table 2 of
Kotowska et al. (2015).

*Rubber PFT development*
For the development of the rubber plant functional type in CLM, our main goal was to
capture the growth characteristics of rubber trees and include a realistic representation of carbon
exports via latex harvest. We adapted the rubber PFT to be partly based on the existing broadleaf
tropical deciduous tree PFT. We modified the phenology scheme and implemented the harvest
export using the field data described above.

*Phenology, Carbon & Nitrogen Allocation and Yield*
First, we considered the default tropical drought deciduous phenology scheme in CLM4.5
(Dahlin et al., 2015) that allows plants to shed their leaves through two alternative mechanisms:
1) stress-deciduous leaf onset/offset switches triggered by a sustained period of dry soil; 2) a
background leaf litterfall rate calculated based on leaf longevity that is not associated with a
specific offset period but occurs over an extended period of time. Leaf onset and offset for the
drought stress of deciduous phenology type (Dahlin et al., 2015) is based on the critical soil
water potential and soil water index accumulator (see Table S1). Preliminary results of the CLM-
rubber development showed that the default drought stress onset/offset mechanism did not
capture well enough the declining trend of LAI of rubber plantation in the dry season. The
determination of leaf shedding for tropical deciduous trees is generally a challenging problem
(Dahlin et al., 2017) and very few studies have looked into this aspect to date (Medlyn et al.,
2016; Xu et al., 2016). Based on the measurements of leaf litterfall (Kotowska et al., 2016), we
incorporated seasonal changes in the leaf life span of rubber in order to improve the background
leaf litterfall mechanism, wherein we set a higher leaf life span value in the wet season than in
the dry season. A calendar month-dependent function of month was used to model leaf life span
("leaf_long" (yrs)). This function computes leaf_long in a step-wise fashion as follows,





$$\text{leaf\_long} = \begin{cases} 2, \text{month} < 4 \\ 1.55, 4 \leq \text{month} \leq 5 \\ 0.55, 6 \leq \text{month} \leq 7 \\ 0.23, 8 \leq \text{month} \leq 9 \\ 4.1, \text{month} > 9 \end{cases} \quad (1)$$
The above implementation was necessary to ensure that the modeled background leaf litterfall
considers the variability in leaf life span.

188   Four to five times in a week, stems of rubber trees are tapped and the latex is harvested

(yield). Previous experimental work showed that tapped rubber trees grew less than untapped
trees (Chantuma et al., 2009; Silpi et al., 2007). Since latex is rich in carbon, this was interpreted
as active carbon allocation to storage in response to tapping (Chantuma et al., 2009; Silpi et al.,
2007). In our model, latex yield is proportional to annual net primary productivity (Kotowska et
al., 2015) and also considered from the partitioning of growth and storage carbon pools. We
included the latter because in the field, latex yield could also result from the storage pools
(Junjittakarn et al., 2012; Sara et al., 2014).

196   To our knowledge, calculation of latex yield from net primary productivity and

calculation of latex yield from the partitioning of growth and storage carbon pool is a new
concept and has not been considered in any of the rubber modeling studies (Kumagai et al.,
2013). Subsequently, we introduced in the CLM-rubber two tapping-related parameters; tap_npp,
the proportion of latex yield taken from net primary productivity and tap_partition, the
proportion of latex yield taken from the partitioning of the growth and storage carbon pools (see
Table S1).

203   CLM4.5 calculates carbon allocated to new growth based on five allometric parameters

that relate allocation between tissue types (Oleson et al., 2013): 1) ratio of new fine root to new
leaf carbon allocation ($a_1$), 2) ratio of new coarse root to new stem carbon allocation ($a_2$), 3) ratio
of new stem to new leaf carbon allocation ($a_3$), 4) ratio of new live wood to new total wood
allocation ($a_4$) and 5) ratio of growth respiration carbon to new growth carbon ($g_1$). CLM4.5 has
a dynamic allocation scheme (Oleson et al., 2013), which includes one dynamic allometric
parameter ($a_3$ as a function of annual NPP). For the drought deciduous tropical PFT, $a_1$, $a_2$, $a_4$ and
$g_1$ are constants ($a_1 = 1$, $a_2 = 0.3$, $a_4 = 0.1$ and $g_1 = 0.3$), whereas $a_3$ is a dynamic parameter
defined by the following equation,
$a_3 = \frac{2.7}{1+e^{-0.004(\text{NPP}_{\text{ann}}-300)}} - 0.4,$              (2)

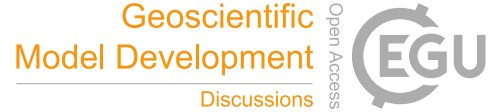



where $NPP_{ann}$ is the annual sum of NPP of the previous year. The above equation for $a_3$ increases
stem allocation relative to leaf when annual NPP increases. We assume that due to tapping, the
ratio of new stem to new leaf carbon allocation would change, thus, for CLM-rubber, we
modified $a_3$ as follows,
$a_3 = \frac{2.7}{1+e^{-0.004((1-tap\_npp)NPP_{ann}-300)}} - 0.4,$ (3)
where tap_npp is the proportion of latex yield taken from annual net primary productivity.
In addition to tapping from NPP, we also considered tapping from the partitioning of
growth and storage carbon pool. We recognized that for all deciduous PFTs, there is a fraction of
allocation that goes into the growth pool (fcur), which is currently set to 0 (unitless) and the
remainder (1- fcur) goes to the storage pool. Subsequently, the deciduous phenology module
either uses the onset growth function or a background growth transfer rate (bgtr; outside of onset
period) to move storage carbon to displayed growth pools. For CLM4.5-rubber, we partition fcur
into three portions; growth (fcur_gr), storage (fcur_st) and tapping (fcur_tap). We assume that
the total fraction of allocation that goes into the growth, storage and tapping pool is 1 (unitless).
Next, we define fcur_tap as a parameter called "tap_partition" (see Table S1).  We also define
the fraction of allocation that goes into the storage as "fcur_st" and set fcur_st = 0.5. Because
fcur_tap and fcur_st are known, we obtain the fraction of allocation that goes into the growth
pool (fcur_gr) as follows,
$fcur\_gr = 1 - (fcur\_tap + fcur\_st)$ (4)
It is important to recognize that in Eq. 4 as fcur_tap increases, fcur_gr decreases. This
trade-off is in line with the notion that tapping limits growth (Chantuma et al., 2009; Silpi et al.,
2007). Given the above allocation parameters ($a_1$, $a_2$, $a_3$, $a_4$ and $g_1$) and carbon to nitrogen ratios
of these tissues: leaf, fineroot, livewood (in stem and coarse root) and deadwood (in stem and
coarse root), which are constants, the total carbon and nitrogen allocation to new growth
($CF_{alloc}$, $gC\ m^{-2}s^{-1}$, and $NF_{alloc}$, $gN\ m^{-2}s^{-1}$, respectively) can be expressed as functions of
new leaf carbon allocation $\left(CF_{GPP,leaf}, gC\ m^{-2}s^{-1}\right)$:
$CF_{alloc} = CF_{GPP,leaf}C_{allom,}$ (5)
$NF_{alloc} = CF_{GPP,leaf}N_{allom,}$
where $C_{allom}$, $N_{allom}$ are the carbon and nitrogen allometry (Oleson et al., 2013). From the
stoichiometric relationship in Eq. 5, the associated carbon allocation flux is:

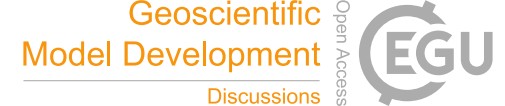

$$\mathrm{CF_{alloc} = NF_{alloc} \frac{C_{allom}}{N_{allom}}} \tag{6}$$
Total allocation to new leaf carbon $\left(CF_{alloc,leaf\_tot}, gCm^{-2}s^{-1}\right)$ is calculated as:
$$\mathrm{CF_{alloc,leaf\_tot} = CF_{alloc} \big/ C_{allom}} \tag{7}$$
In CLM4.5, there are two carbon pools associated with each plant tissue: 1) growth and
2) storage. The carbon pools that represent growth include carbon in leaf (leafc), carbon in fine
roots (frootc), carbon in live stem (livestemc), carbon in dead stem (deadstemc), carbon in live
coarse roots (livecrootc) and carbon in dead coarse roots (deadcrootc). The carbon pools that
represent storage have a suffix "_storage" and include leafc_storage, frootc_storage,
livestemc_storage, deadstemc_storage, livecrootc_storage and deadcrootc_storage. In CLM4.5,
the carbon allocation fluxes have a prefix "cpool_to_".
For CLM-rubber, we made changes to all of the above carbon pools, we show below the
key carbon allocation fluxes for CLM-rubber in the leaf, fineroot and tapping pools. Given
$CF_{alloc,leaf\_tot}$, fcur_gr, fcur_st and fcur_tap, the allocation fluxes of carbon to growth and
storage pools for the various tissue types can be calculated as follows,
$\mathrm{cpool\_to\_leafc = CF_{alloc,leaf\_tot} * fcur\_gr,}$ $\tag{8}$
$\mathrm{cpool\_to\_leafc\_storage = CF_{alloc,leaf\_tot} * fcur\_st,}$
$\mathrm{cpool\_to\_frootc = CF_{alloc,leaf\_tot} * a_1 * fcur\_gr,}$
$\mathrm{cpool\_to\_frootc\_storage = CF_{alloc,leaf\_tot} * a_1 * fcur\_st}$
The carbon flux of the latex yield is the sum of yield from net primary productivity and
partitioning pool, which is calculated as follows,
$\mathrm{cpool\_to\_tappingc = tap\_npp * NPP\_ann * \frac{1}{3600*24*365} + CF_{alloc,leaf\_tot} * fcur\_tap}$ $\tag{9}$
Besides the new tapping mechanism, one of the major differences between CLM4.5 for
tropical deciduous PFT and CLM-rubber with respect to the above carbon allocation fluxes is
that CLM4.5 does not have a carbon export flux for latex yield. Further, CLM4.5 used a fixed
"fcur" term in all of the equation 8 to partition carbon fluxes to growth pools and to storage pools
associated with each tissue type. In contrast, CLM-rubber partitions allocation fluxes to the new
tapping pools as well as growth and storage, which are defined above in Eq. 4.
The nitrogen pools follow the stoichiometric relationship with carbon pools. The nitrogen
pools for the growth include nitrogen in leaf (leafn), nitrogen in fine roots (frootn), nitrogen in





live stem (livestemn), nitrogen in dead stem (deadstemn), nitrogen in live coarse roots
(livecrootn) and nitrogen in dead coarse roots (deadcrootn). Similar to the carbon pools, the
nitrogen pools also have corresponding storage pools and displayed growth pools The
corresponding nitrogen allocation fluxes are calculated as ratios of carbon allocation fluxes using
the inverse of respective carbon to nitrogen ratios of different tissue types.

In the CLM-rubber, tapping of rubber trees started at the age of six years. The model

updates the ratio of carbon to nitrogen of latex yield every half hour.

*Initial model simulations*

To mimic the vegetation and soil state prior to rubber plantation, a Tropical Evergreen

forest PFT was first spun-up and run until 1997 using the standard procedures of CLM4.5 spin-
ups (Fan et al., 2015; Koven et al., 2013). We used the Tropical Evergreen PFT for the spin-up
because we assumed that the natural vegetation prior to land-use change was evergreen. A
comparison of the modeled above ground biomass and net primary productivity of the spun-up
phase with the observed above ground biomass and net primary productivity of tropical
evergreen forests at our site (Kotowska et al., 2015) showed that these matched reasonably well
(see Figure S1). Following the spin-up phase, a clear-cut in 1998 was simulated by setting the
above ground carbon and nitrogen pools to zero.

Second, using the site-level measurements on soil texture (Allen et al., 2015) and climate

data for 2013 (Meijide et al., 2018) at the Harapan landscape, a rubber plantation simulation was
performed from 1998 to 2014. First, we used the default parametrization of stress deciduous
tropical PFT of CLM4.5, but the generic model performance was poor relative to some of the
measurements (see below for details). Hence, we performed a model calibration exercise using
data collected specifically for rubber plantations.

*Observational data for model calibration*

We briefly outline all of the measured data that we used for parametrization and

calibrating the CLM-rubber in the Harapan landscape. Except for fine root biomass, all other
data, which consist of carbon pools and net primary productivity in above- and below ground
tree biomass at our plots are adapted from Kotowska et al. (2015). We used measurements of
fine root biomass from Kurniawan et al. (2018) because it was measured down to a depth of 100



cm. This data-set is not from a different site and was measured across all eight rubber plots. The
overall measurement campaign for the rubber inventory data spanned from August 2012 to
March 2014. Leaf litterfall of data were collected using 16 litter traps (placed in a random grid)
at each of the eight  plots (see Kotowska et al., 2016 for details). Litter was collected at monthly
intervals from March 2013 to April 2014.

Soil respiration was measured using vented static chamber, to which four permanent

chamber bases were placed randomly at each of the eight plots. Concurrent to soil respiration
measurement, soil moisture was measured by the gravimetric method using four samples taken
near to the chambers within a depth of 5 cm. Both soil respiration and moisture content were
measured monthly at all plots from December 2012 to December 2013 (Hassler et al., 2015).

Rubber tree water use ('transpiration') was measured using two commonly applied sap

flux techniques, the thermal dissipation probe (TDP) method (Granier, 1985) and the heat field
deformation (HFD) method (Nadezhdina et al., 2012). Two TDPs per tree yielded averages of
sap flux density in the outer xylem (0-2.5 cm) for each sample tree. The HFD method, with
multiple measurement points from 0-8 cm into the xylem, yielded typical radial sap flux profiles
for rubber trees and thus allowed the calculation of cross-sectional water conductive areas.
Combining the output of the two methods allowed us to calculate water use rates of the six
sample trees per plot, which was further extrapolated to stand transpiration (using the tree
density and diameter distribution from Kotowska et al., (2015)) (Niu et al., 2017).

*Model calibration steps & resource limitations*

During the initial model – measurement comparison, we noted several discrepancies

between the model and measurements. Compared to the modeled values, the measured
transpiration and leaf area index were substantially lower while soil respiration was higher. To
minimize the mismatch between the model and measurements, we decided to calibrate the
model. Due to the long computing time required to run the CLM model (from 1998 to 2014), in
this study, we used a simple calibration method (Fan et al., 2015; Rahul et al., 2014) as opposed
to more complex methods such as Monte Carlo Markov Chain approaches (Ali et al., 2016).

Our initial model calibration step involved obtaining a "realistic" seasonal dynamics of

leaf area index. Although we did not have the seasonal data on leaf area index, our educated
guess (as well as through "pers. comm.") indicates that the seasonal dynamics of leaf area index





would be relatively "smooth" with a depth of the dip not so large in the dry season, that is, it will
have something like a "brevi-deciduous" phenology. We do not expect the leaf area index in the
dry season to decrease suddenly with a strong intense as observed in rubber plantations from
other sub-tropical regions (see Fig.2; Giambelluca et al., 2016). We increased the critical value
of the soil water potential (from -2 MPa to -0.5 MPa) to trigger leaf shedding in the model. In
this case, the seasonal dynamics of the modeled leaf area index resulted in a sudden decrease in
leaf area index with a narrow depth of the dip – a seasonal trend of leaf area index that we do not
expect at our study sites.

In CLM4.5, soil water influences stomatal conductance directly by multiplying the

minimum conductance by a soil water stress function $\beta_t$ and also indirectly through net
photosynthesis (Oleson et al., 2013). The latter effect is achieved by multiplying the maximum
carboxylation rate ($V_{cmax}$) and dark respiration ($R_d$) by $\beta_t$. The function $\beta_t$ ranges from one
when the soil is wet to near zero when the soil is dry and depends on the soil water potential of
each soil layer, the root distribution of the plant functional type, and a plant-dependent response
to soil water stress
$\beta_t = \sum_i w_i r_i$                                                                        (10)
where $w_i$ is a plant wilting factor for layer $i$ and $r_i$ is the fraction of roots in layer $i$. The plant
wilting factor $w_i$ is
$w_i = \begin{cases} \frac{\varphi_c - \varphi_i}{\varphi_c - \varphi_o} \left( \frac{\theta_{sat,i} - \theta_{ice,i}}{\theta_{sat,i}} \right) \le 1, T_i > T_f - 2 \text{ and } \theta_{liq,i} > 0 \\ 0, T_i \le T_f - 2 \text{ or } \theta_{liq,i} \le 0 \end{cases}$                (11)
where $\varphi_i$ is the soil water matric potential (mm) and $\varphi_c$ and $\varphi_o$ are the soil water potential (mm)
when stomata are fully closed or fully open (respectively). The term in brackets scales $w_i$ by the
ratio of the effective porosity (accounting for the ice fraction; $\theta_{sat} - \theta_{ice}$) relative to the total
porosity.

To induce stomatal closure via soil water in CLM-rubber, we increased the sensitivity of

stomata to soil water stress (Verhoef and Egea, 2014) by modifying the default soil water
potential for drought deciduous tropical PFT in the model for stomatal opening "smpso = -17500
mm" to  "smpso = -8750 mm" and stomatal closing "smpsc = -112000 mm" to "smpsc = -56000
mm". These changes are equivalent to modifying stomatal opening from -0.34 MPa to -0.17 MPa
and full closure from -2.19 MPa to -1.09 MPa. The above two changes are within the range of
plausibility, if we consider rubber trees to be sensitive to drought. The values of soil water



potential for stomatal opening and full closure in CLM depend on plant functional type. The
default values for stomatal opening of PFTs range from -0.35 to -0.83 MPa while for full closure
Oleson et al. (2013) quote values ranging between -2.24 and -4.28 MPa. These stomatal opening
and full closure values in CLM are known to vary a lot by species and are based on White et al.

(2000).

Rubber plantations at our sites are known to have low soil nitrogen availability and are

not fertilized. In particular, the rubber plantations have low gross nitrogen mineralization rate,
microbial nitrogen and mineral nitrogen (Allen et al., 2015; Hassler et al., 2015) and therefore,
growth and productivity of our rubber plantations could be limited by nitrogen and possibly by
other nutrients e.g. low phosphorus and base saturation (Allen et al., 2016). These were
attributed to the fact that our studied  plantations were on highly weathered acrisol soils (which
have inherently low levels of extractable phosphorus and exchangeable base cations) and were
not fertilized for two to five years prior to the start of our field measurements in 2012 (Allen et
al., 2015; Hassler et al., 2015; Kurniawan et al., 2018). In an attempt to capture the magnitude of
the relatively low leaf area index and low transpiration, we made the following change based on
the idea that if nutrients are limiting in the soil, then in real ecosystems roots will have to pay a
cost. In this version of the model, we assume that maintenance respiration of fine roots is high to
pay for nitrogen uptake, so the base rate of maintenance respiration was increased by 50% for the
fine roots in line with Doughty et al. (2018). In CLM4.5, the base rate of maintenance respiration
per unit nitrogen content is fixed for all tissues (leaf, livestem, livecroot and fineroot) and is
defined as $MR_{base} = 2.525\ e^{-6} gC\ gN^{-1} s^{-1}$. For CLM-rubber, we set $MR_{base}$ to
$3.7875\ e^{-6} gC\ gN^{-1} s^{-1}$ when we calculate the maintenance respiration for fineroots. This
change to represent local nutrient limitation made the model predict a relatively high soil
respiration rate (sum of autotrophic and heterotrophic respiration), thus reducing net primary
productivity and lowering leaf area index.

Preliminary calibration results showed that the modeled soil respiration still

underestimated the measured soil respiration by approximately 25%. To improve this, we
increased the growth respiration of fine roots, which is currently fixed and set as 0.3 for tropical
deciduous PFTs in CLM4.5 by a factor of 3 for CLM-rubber. This is a relatively large change.
There is one reason to support this increase on growth respiration of fine roots. On average, these
rubber plantations lose 20% of the original organic carbon in the soil after 4 years from forest





conversion (van Straaten et al., 2015), yet soil respiration was comparable to that of the reference
forest (Hassler et al., 2015). These findings suggest that the proportion of heterotrophic
respiration would be lower than the contribution of autotrophic respiration to the soil respiration.
The decreases in available nitrogen, extractable phosphorus and base saturation (Allen et al.,
2016; Allen et al., 2015) suggest that there may be strong competition for phosphorus such that
trees have to allocate more carbon for their root growth and root–mycorrhizal system to obtain
these nutrients (Fisher et al., 2016; Shi et al., 2016).

In the model – measurement comparison for soil organic carbon, CLM-rubber initally

predicted only 9% decline in soil carbon for rubber plantation since clear-cut; however, a study
by van Straaten et al. (2015), who conducted soil carbon measurements on heavily weathered
soils for rubber plantations in Jambi and showed that on average, rubber plantations have 20%
lower soil carbon stocks than forests. To increase the modeled decline in soil carbon, we
increased the value of Q10 ("the increase of soil respiration per 10°C increase in temperature")
of soil temperature, from 1.5 to 3, on the grounds that rubber plantations at our study sites are
0.5°C hotter than forests (Meijide et al., 2018).

*Model validation in the Bukit Duabelas landscape*

Using the soil texture measurements from the Bukit Duabelas landscape (Allen et al.,

2015), a model spin-up was performed till year 2002. The spin-up for the model validation was
carried out in the same way as the spin-up for the model calibration. Then a clear-cut was
introduced in 2003. Using climate data from 2013 (Meijide et al., 2018), we performed a
simulation from year 2003 till 2014 by recycling the climate data every year. We used the same
rubber PFT parameterization as obtained for the Harapan landscape except the tap_npp
parameter. The latter was adjusted because (1) the proportion of measured latex yield relative to
measured NPP in the Bukit Duabelas landscape is 10% higher than that in the Harapan landscape
(Kotowska et al., 2015), and (2) the amount of measured latex yield was also higher in the Bukit
Duabelas landscape than the Harapan landscape (Kotowska et al., 2015), although it was not
statistically different. To save space, we include figures for the model calibration in the main
manuscript and put figures for the model validation in the supplementary section.

*Hypothesis testing*



Understanding tropical droughts is important because it affects the growth and mortality
of trees (e.g. Bretfeld et al., 2018; Moser et al., 2014; Phillips et al., 2010). Sometimes drought
can be really hard on forests, where too much heat, low humidity and not enough water can
drastically alter which trees survive (Lewis et al., 2011; Rowland et al., 2015). In the future,
drought is projected to increase (Jiménez-Muñoz et al., 2016; Neelin et al., 2006); however, our
ability to predict how future dry conditions would impact rubber tree productivity and yield is
limited. Therefore, we used CLM-rubber to investigate the impacts of future drought on rubber
yield. We expected drought to reduce the productivity of rubber trees in addition to the latex
yield. We focused on five low rainfall scenarios; two simulations assumed low rainfall to occur
throughout the year and so these simulations had 20%, 50% lower precipitation than the default
precipitation;  the other two simulations assumed low rainfall to occur with the extended dry
season and so precipitation from April to October was reduced by 30%, 50%, in these
simulations; and the final simulation considered shorter dry season but with intense drought so in
this simulation precipitation from 8$^{th}$ May to 12$^{th}$ September was reduced by 50%. We then
performed six simulations of 10-year period from 2015 to 2024; first using the present-day
climate data, and then for the other five simulations, we used the climate data that imposed
drying.

*Leaf life span and specific leaf area*
In the current version of the CLM-rubber, specific leaf area (SLA) is fixed; which is the
case for many land surface models. It has been suggested that SLA could decline with leaf age
e.g. due to leaf economics. We do not have temporal data on SLA for rubber. Because we
developed a rubber model where we included the temporal changes in leaf life span for better
model fit to the leaf litterfall data, we decided to investigate the effect of a dynamic SLA on the
modeled photosynthesis of the rubber at the leaf-level. In CLM4.5 as well as in CLM-rubber,
SLA is referred to as "slatop" – the SLA at the top of the canopy. To have a dynamic SLA, we
let SLA to be low when the leaf life span is relatively high and SLA to be high when leaf life
span is relatively low. The leaf life span is high in the wet than the dry season. We used a
calendar month-dependent function to model the dynamics of slatop:
$$\text{slatop} = \begin{cases} 0.0197, \text{month} \leq 5 \\ 0.024, 6 \leq \text{month} \leq 9 \\ 0.0197, \text{month} > 9 \end{cases} \qquad (12)$$



In Eq. 12, slatop is low in the wet than the dry season, where we reduced slatop by 18% in the
wet season.

*Comparison with other models and locations*

We do not have a flux-net tower in the rubber plantations in Jambi, Indonesia. However,

the CLM-rubber model has been calibrated to carbon and water flux related variables for rubber
plantations at Jambi, Indonesia. Therefore, we think that the modeled estimates of carbon and
water fluxes of CLM-rubber at Jambi, Indonesia can be considered as a "proxy" of measured
fluxes of rubber plantation in Jambi, Indonesia. Thus, we have an opportunity now to compare
modeled estimates of carbon and water fluxes of CLM-rubber in Jambi, Indonesia with
measurements of fluxes from two  rubber plantations at other locations in the Southeast Asia
(Giambelluca et al., 2016). To check the robustness of the CLM-rubber in prediction of the water
fluxes, we compared its modeled water fluxes with the predicted values from a soil water model
(Kurniawan et al., 2018), that is parameterized with the site-specific soil physical and
hydrological parameters from our studied plots. Finally, to identify the relative ranking of the
above-ground carbon stock of rubber plantations, we compared the measured and modeled
estimates of carbon from our site in Jambi province, Indonesia with measurements from China,
Africa and Brazil (Kotowska et al., 2015; Wauters et al., 2008; Yang et al., 2016).

**Results**
*Dynamics of carbon use*

CLM-rubber was able to simulate the dynamics of net primary productivity (Figure 2; a),

above-ground biomass (Figure 2; b) and total soil organic carbon (Figure 2; c) of the rubber
plantation in the Harapan landscape. The modeled biomass of fine roots and the annual latex
yield were also within the measured range (Figure 3; a, b). When validated in the Bukit Duabelas
landscape, the modeled net primary productivity (Figure S2; a) and above ground biomass
(Figure S2; b) were quite close to the measurements. The modeled biomass of the fine roots and
the annual latex yield were much closer to the measurements in the model validation case
(Figure S3; a, b) than the model calibration case (Figure 3; a, b).





Despite the large variability across plots for the measured values, CLM-rubber captured
the seasonal dynamics of the leaf litterfall (Figure 4a) far better than the seasonal trends of soil
respiration (Figure 4b) and soil moisture (Figure 4c) in the Harapan landscape. For leaf area
index, the measured values in 2018 were below our simulated values for 2014 (Figure 4d); there
may be also an inter-annual variability of leaf area index, aside from seasonal variability, since
for the leaf litterfall to be captured well by the CLM-rubber the LAI must be predicted
reasonably well for 2014. The modeled seasonal patterns of carbon and water dynamics at the
Bukit Duabelas landscape (Figure S4) were similar with those at the Harapan landscape.

*Dynamics of water use*
The calibrated model in the Harapan landscape was close to the pattern and magnitude of
the measured diel transpiration in a dry (Figure 5; a) and wet month (Figure 5; b). The modeled
and measured diel courses of transpiration were characterized by relatively low hourly maxima
($< 0.25$ mm h$^{-1}$; Figure 5). The model had an early onset as well as an early offset of
transpiration than the measurements (Figure 5; a, b). This is consistent with the diurnal effects
that nitrogen limitation is known to have in CLM4.5 (see Fig.1; Ghimire et al., 2016). The model
successfully predicted the average transpiration of a 2-year and 5-year old rubber plantations
(Figure 6; a, b).
The validated results in the Bukit Duabelas landscape showed that the diel trends of
predicted and measured transpiration were quite similar (Figure S5; a-c) to those in the Harapan
landscape. The model captured the long-term seasonal trends of transpiration well (Figure S6),
except for a minor discrepancy for a few weeks in June, where there was some period of partial
leaf shedding. The magnitude of the modeled transpiration was also quite close to the
measurements (Figure S6).

*Leaf life span and specific leaf area*
Since in the CLM-rubber the trees have a drought-deciduous leaf phenology, we
investigated the effect of fixed versus dynamic specific leaf area (Figure 7; a, b) and found that
the mass-based photosynthesis of rubber leaves had a stronger dependence on leaf life span when
the specific leaf area is dynamic (a higher $r^2$ value; Figure 7; b) rather than fixed. Interestingly,





the model predicted that a higher mass-based photosynthesis of the rubber leaf can be associated
with a lower leaf life span - this is a proposition that cannot be drawn if the specific leaf area is
fixed (i.e. Figure 7; a). This finding suggests that long-lived rubber leaves could have a low
mass-based photosynthesis, and that rubber plants could spend carbon in the construction of
other tissues such as those associated with protection against insects or prevention of leaf
diseases.

*Model Projection*
CLM-rubber predicted reduced yield in response to different drought scenarios as the
intensity of drought increased (Figure 8). Modeled yield tended to have a non-linear relationship
with soil moisture. CLM-rubber predicted up to a 18% reduction in yield when the intensity and
duration of drought was largest (Figure 8). Currently, we do not have field data to confirm the
magnitude of the effect of drought on yield, predicted by the model.

*Comparison with other sites and models*
CLM-rubber predicted a lower carbon uptake in the wettest month for Jambi, Indonesia
compared to measurements of a similar plantation from Thailand (Table 1). The model also
predicted a lower carbon release in the driest month for Jambi than a similar plantation from
Thailand (Table 1). It should be pointed out that the rubber plantations in Jambi were unfertilized
in the recent years and are on highly weathered acrisol soils with low fertility while the rubber
plantation in Thailand are highly fertilized. The CLM-rubber carbon fluxes suggest that rubber
plantations from our sites are unlikely to have high carbon uptake or releases at the ecosystem
scale compared to rubber plantations from other parts of Asia because the rubber plantations
from our sites are not fertilized and have low leaf area index. At the ecosystem scale, CLM-
rubber predicted a lower annual evapotranspiration and higher sensible heat fluxes from our sites
than a similar plantation from Thailand (Table 1). These results indicate that rubber plantations
from our sites are likely to have a high canopy openness than rubber plantations from other parts
of Asia.
The comparison of water fluxes of CLM-rubber with a site-parameterized soil water
model showed that CLM-rubber can predict the water fluxes reasonably well (Table 2). When
comparing the carbon stocks of Jambi, Indonesia with other tropical countries, we found that the





above-ground biomass of rubber plantations was mostly similar except for a plantation from
Africa (Table 3).

**Discussion**
*Phenology and Carbon & Nitrogen Allocation*

The seasonality observed in the empirical leaf litterfall data represented a challenge for

the development of the CLM-rubber. During CLM-rubber development, we realized that the
version of CLM-rubber that did not consider the temporal changes in leaf life span (that had
fixed leaf life span as 1 year) failed to capture the seasonality of leaf litterfall. We have
demonstrated in this study the importance of temporal changes in leaf life span. Seasonal data on
leaf life span and leaf area index for rubber trees will be invaluable to capture well the carbon
cycle of CLM-rubber. Similar data sets for tropical deciduous trees should be collected, which
currently are rare (Dahlin et al., 2017). Our study suggests that land surface models should not
use fixed leaf life span for simulating carbon dynamics of tropical deciduous PFTs if the focus of
the study is examining seasonal pattern.

Radiation intensity has been suggested to play an important role in the onset of rubber

leaves for the sub-tropics (Hoong-Yeet, 2007). At this stage, we did not integrate radiation
intensity to trigger the onset of rubber leaves in the CLM-rubber because we do not have
sufficient phenology data. Currently, the trigger for leaf onset in CLM-rubber is based on soil
water potential. The carbon cycle of CLM-rubber can therefore be further improved by
examining possible controls on leaf shedding and flushing in rubber and their interactions, e.g.
soil water potential and radiation intensity.

The seemingly higher latex yield of rubber plantation in the Bukit Duabelas landscape

compared to the Harapan landscape (despite being five years younger than Harapan) could be
due to differences in management practices between the two landscapes (e.g. tapping frequency,
planting density; Kotowska et al., 2015) and/or differences in soil texture, which influences
differences in fertility (Allen et al., 2016; Allen et al., 2015; Kurniawan et al., 2018). The change
in Q10 value of soil temperature enabled CLM-rubber to predict a 16% decline in soil carbon
since clear-cut – a finding that is similar to a study by van Straaten et al. (2015). Indeed, in a





recent study, Meyer et al. (2018) have shown that Q10 has a lot of variability across PFTs,
ranging from 1.25 to 2.75.

*Low transpiration rates*

The inclusion of increased sensitivity of stomata to soil water stress and the enhancement

of growth and maintenance respiration of fine roots in response to soil nutrient limitation enabled
CLM-rubber to capture the magnitude of transpiration and leaf area index; however, the model
had an earlier onset of diel transpiration (Figure 5; a, b) as well as an earlier offset of
transpiration (Figure 5; a, b) than the measurements. The early onset of modeled transpiration
around 8 am can be explained by the relatively high radiation (Figure S7; a) while the early
offset of modeled transpiration around 6 pm (Figure 5; a) can be related to the absence of the
stem water storage term in the model. The sap flow measurements could also have uncertainties
due to their set-up. The sensors were inserted in the tree trunk at about 2 m height. Above this
height, there could be considerable water storage in the plant. Early in the morning, transpiration
may make use of this water storage – as indicated by the modeled transpiration. In the evening,
the plant water storage above the sap flow sensors may be refilled, and thus water flow at the
trunk is measured. Another source of error in the measurements of transpiration can be related to
the fact that there were only 5 sap flow sensors, which were then used to upscale transpiration to
the canopy-level.

CLM-rubber showed that rubber plantations can exhibit two peaks of leaf photosynthesis

during a day (Figure S8; a), which could be due to the existence of optimal climatic conditions
operating at multiple times within a day (Figure S7; a-c). Although absorbed PAR remained
relatively high around noon time (Figure S8; b), the modeled leaf photosynthesis declined due to
limitations in soil water and stomatal conductance (Figure S8; a). The model simulated the long-
term dynamics of transpiration close to  measurements (Niu et al., 2017). Our results are not
consistent with reports speculating that rubber trees could be large carbon sinks (Kumagai et al.,
2013) and behave as 'water pumps' (Tan et al., 2011; Ziegler et al., 2012).

Other factors such as carbon economy, plant health and soil degradation (Sitorus and

Pravitasari, 2017) could also constrain the productivity and water use of rubber at our studied
sites. CLM-rubber clearly provides additional opportunities to test hypotheses of the effects of




climate scenarios, management practices to alleviate nutrient limitations or their combinations on
carbon economy of rubber plantations.

*Dynamic traits & uncertainties in leaf area index*
From this CLM-rubber development, we can derive suggestions for improving current
land surface models. While the carbon, water and nutrient cycles in land surface models have
improved considerably, the development of trees from seedlings to mature growth phases is less
well represented (Fisher et al., 2018). Our model clearly demonstrates that some of the basic
plant functional traits, e.g. leaf life span, even specific leaf area that are currently considered
fixed parameters in land surface models, need a dynamic seasonal component (Girardin et al.,
2016; Lopes et al., 2016; Wu et al., 2016). This may further apply for longer-term dynamics, e.g.
with regards to different growth phases. Follow-up research is needed to align seasonal and
growth phase-related plant traits, e.g. leaf life span, fruiting of rubber trees, and leaf area index.
From a theoretical point of view, very young and old leaves are unlikely to have a mass-
based photosynthetic rate as high as that of fully expanded mature leaves. Broadly, this finding
has some support from tropical studies (Albert et al., 2018) but needs to be evaluated for rubber.
The fact that CLM-rubber did not capture the magnitude of the measured leaf area index
in 2018 (measured with a LAI 2000 measurements, LiCor Biosciences Inc.) can be due to large
variability in climatic factors, such as flux density of photosynthetically active radiation as well
as the time of measurement (Cotter et al., 2017). We also obtained leaf area index for year 2014
from MODIS satellite on clear sky days for the studied rubber plantations. The MODIS leaf area
index was as high as 4 $m^2$ $m^{-2}$, which is similar to the predictions of CLM-rubber.

*Opportunities for CLM-rubber*
As CLM-rubber predicted reasonably well the carbon and water use, we think that the
current model can be used for larger-scale simulations within Indonesia, in particular, the
lowland areas with mineral soils of Jambi province by incorporating in the prediction soil texture
as the surrogate variable for the control of soil fertility and soil moisture. CLM-rubber can aid in
science-based management and policy recommendations, as the model can be applied to
scenarios of soil management intensities, climate variations, and policy-driven land-use change



projections. CLM-rubber model can also be applied to rubber plantations in other regions in
Southeast Asia but it will require validation against measured carbon, water and energy flux data
from the Asia flux community (Giambelluca et al., 2016; Kumagai et al., 2013; Tan et al., 2011).

Plot-level simulations can potentially be performed for so called jungle rubber plantations

(Feintrenie and Levang, 2009; Gouyon et al., 1993), where the rubber and the trees from the
natural tropical forest coexist. Here, we can use the newly developed CLM-FATES model,
which has a demographic component that considers processes such as height-structured
vegetation and competition between individuals for light (Fisher et al., 2015). In Jambi province,
jungle rubber represents a smallholder rubber agroforestry system, which is established by
planting rubber trees into (often previously logged) rainforests. Similar measured data used in the
current study exists for eight jungle rubber plots differing in soil texture, nutrient levels and
water characteristics. The abundance of natural and rubber trees need to be incorporated in the
model and then carbon and water cycles can be investigated.

Additional experimental data in the dry season on leaf aging and fruiting of rubber should

be collected to investigate if rubber plants take advantage of the high light availability, while
coping with high atmospheric water demand and low water supply. These empirical data can be
an indicator of adaptive strategies of how rubber plants optimize reproduction and resource
acquisition.

**Final Remarks**

Incorporating a dynamic leaf life span enabled CLM-rubber to better capture the

seasonality of leaf litterfall. Increased sensitivity of stomata to soil water stress and the
enhancement of growth and maintenance respiration of fine roots in response to soil nutrient
limitation enabled CLM-rubber to capture the magnitude of transpiration and leaf area index.
Our results show that rubber plantations in Jambi are less likely to have similarly high carbon
fluxes and water use compared to highly fertilized rubber plantations from other parts of South-
east Asia such as those from Thailand and Cambodia.



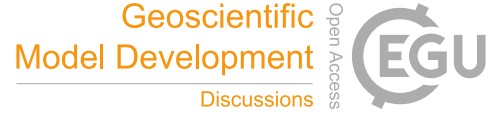



**Code & Data Availability**


Code is available on GitHub (https://github.com/ashehad/CLM4.5_rubber_v1/tree/master/codes)
and data used in this paper can be found in this repository
(https://github.com/ashehad/CLM4.5_rubber_v1/tree/master/data/measured_data_for_model_cali
bration).

**Acknowledgements**
We gratefully acknowledge financial supports from Deutsche Forschungsgemeinschaft (DFG) in
the framework of the collaborative German-Indonesian research project CRC990 in subproject
A07. We thank Syahrul Kurniawan for running the soil-water model and Aiyen Tjoa from the
University of Jambi (UNJA), Jambi, Indonesia for taking canopy pictures of rubber plantations
during the dry season. Katie Dagon from National Center for Atmospheric Research is thanked
for discussing the hydrological processes in CLM4.5. We also thank George Ofori Ankomah
from the University of Goettingen for measuring the leaf area index at the studied rubber
plantations. Finally, we thank the village leaders, PT REKI and Bukit Duabelas National Park for
allowing us to conduct our research on their land.

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



**Figures**

**Figure 1** Illustration of the original and modified structure and functions of CLM4.5 for incorporating the rubber plant functional type (PFT). The original functions in CLM4.5 are represented in black while the new rubber PFT in CLM4.5 are represented in red, which includes changes to phenology, allocation of carbon and nitrogen, and harvest algorithm.

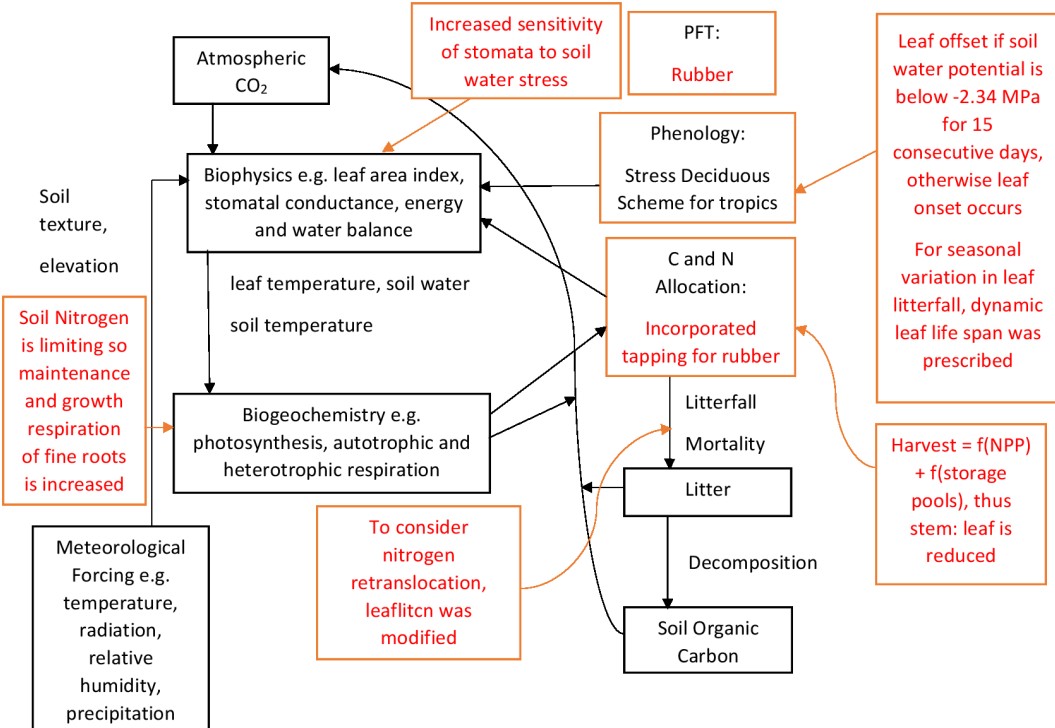




**Figure 2** Temporal trends of annual net primary productivity (NPP; kg C m$^{-2}$ yr$^{-1}$) annual above
ground biomass (AGB; kg m$^{-2}$), and total soil organic carbon content up to 3 m (TSOC; kg m$^{-2}$)
of rubber, simulated by CLM-rubber following clear-cut in 2001 in the Harapan landscape.
Measured NPP, AGB and TSOC (lines are standard errors, n = 4 plots) are indicated for 2014.

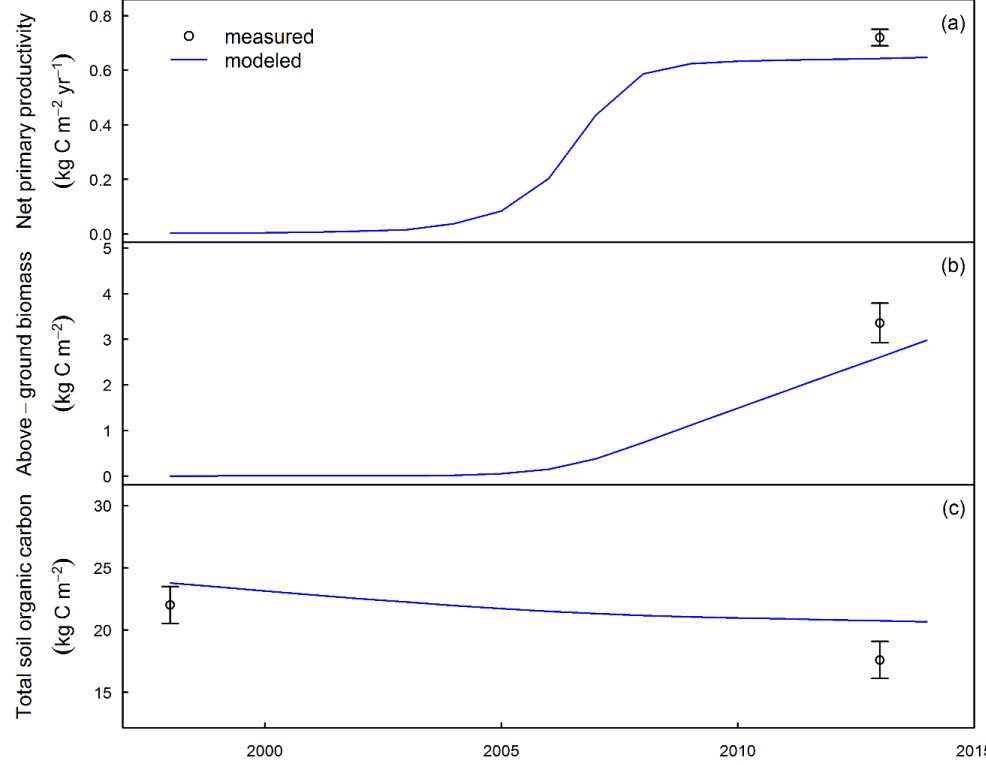









**Figure 3** Measured (lines are standard error, n = 4 plots) and CLM-simulated fine root biomass
(a) and annual latex yield (b) of rubber plantation in 2013 in the Harapan landscape.

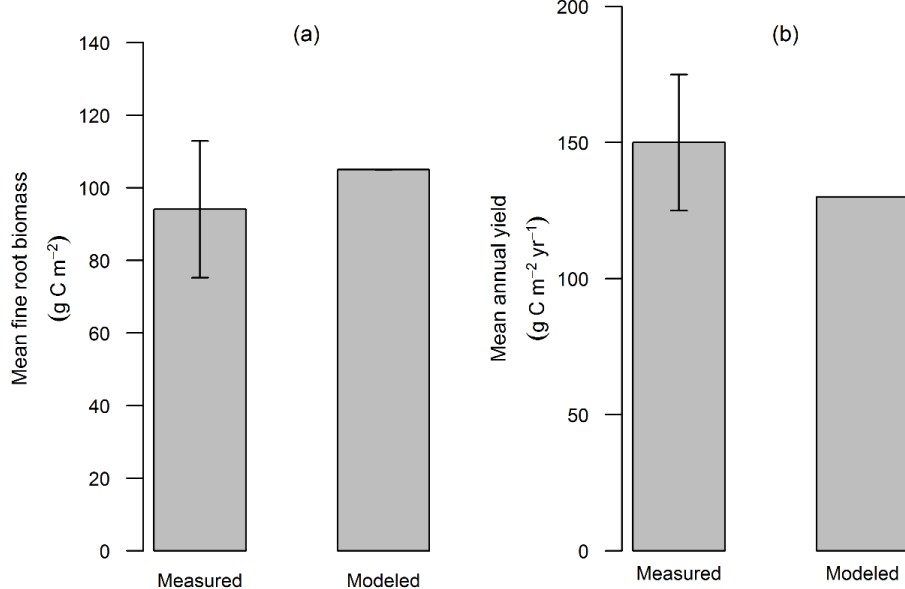



**Figure 4** Monthly trends of leaf litter fall ((a); g C m$^{-2}$ yr$^{-1}$), soil respiration ((b); kg C m$^{-2}$ yr$^{-1}$),
soil moisture up to 5 cm ((c); m$^3$ m$^{-3}$) and leaf area index ((d); m$^2$ m$^{-2}$) of rubber plants simulated
by CLM-rubber (blue line) and observed values (open circles) during the mature phase of growth
of rubber. The leaf area index (LAI) was measured in 2018. In April, LAI was measured in only
one plot whereas in May, LAI was measured across all four plots. The vertical line in April is the
standard error across the first plot while the vertical line in May is the standard error across all
four plots.

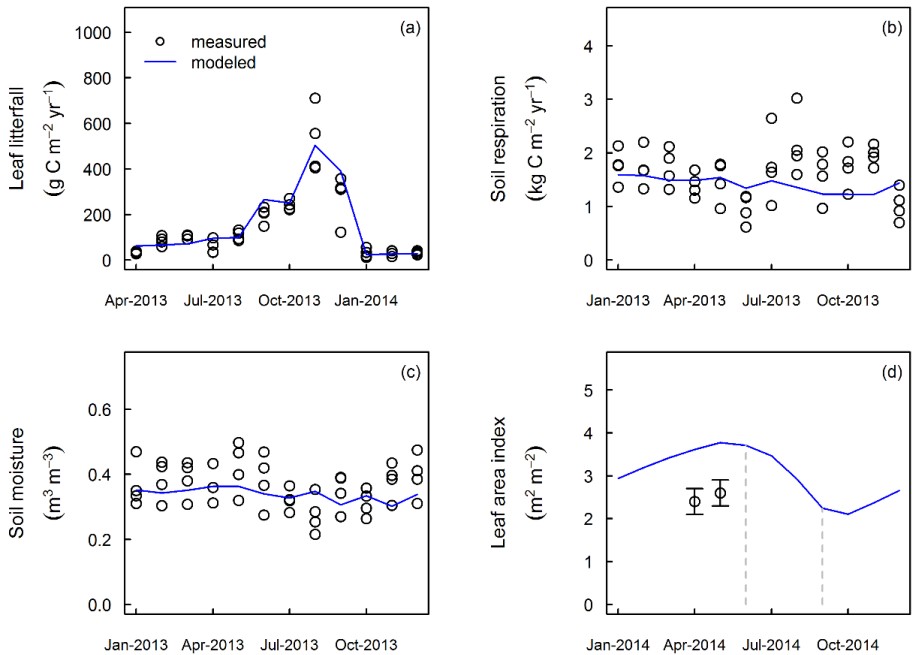








**Figure 5** Measured and modeled diel transpiration (mm hr$^{-1}$) of rubber averaged over June (dry
month) and December (wet month) in the Harapan landscape in 2013.

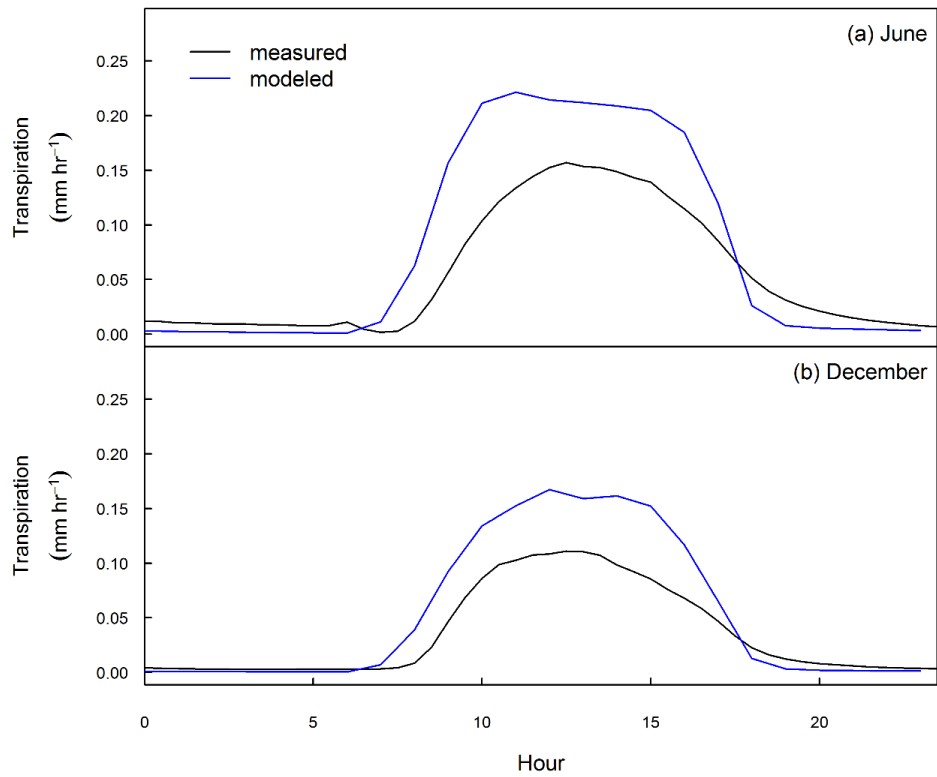






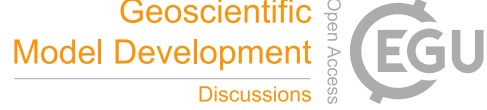

**Figure 6** Measured and CLM-simulated transpiration of (a) a 2-year old rubber over December, 2013 and (b) a 5-year old rubber over January, 2014 in the Harapan landscape. The bars and the lines are means and standard errors, respectively, over half-hourly data of each month.

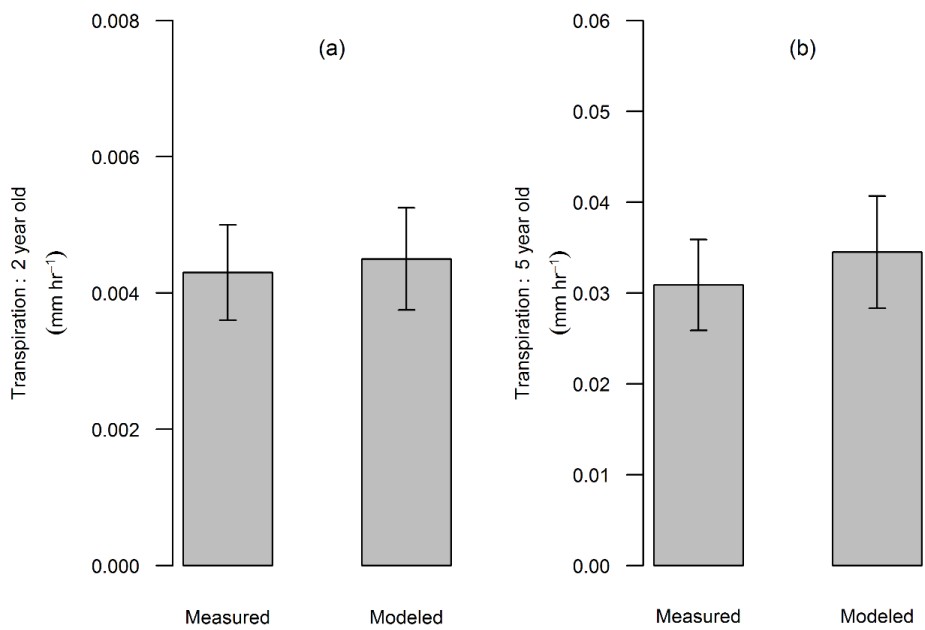







**Figure 7** Effect of fixed specific leaf area (SLA) (a) versus dynamic SLA (b) on the CLM-
simulated photosynthesis of rubber leaves, expressed on leaf mass basis, as a function of leaf life
span. Each data point corresponds to the monthly value, which is an average of the peak
photosynthesis between 10 am and 2 pm. The data points corresponding to the lowest leaf life
span belong to the dry season while those at mean leaf life span correspond to the period before
the leaf fall. The data points corresponding to the highest leaf life span correspond to the period
after the leaf fall. The blue dashed line is the best fit with the goodness of fit indicated by the $r^2$
value.

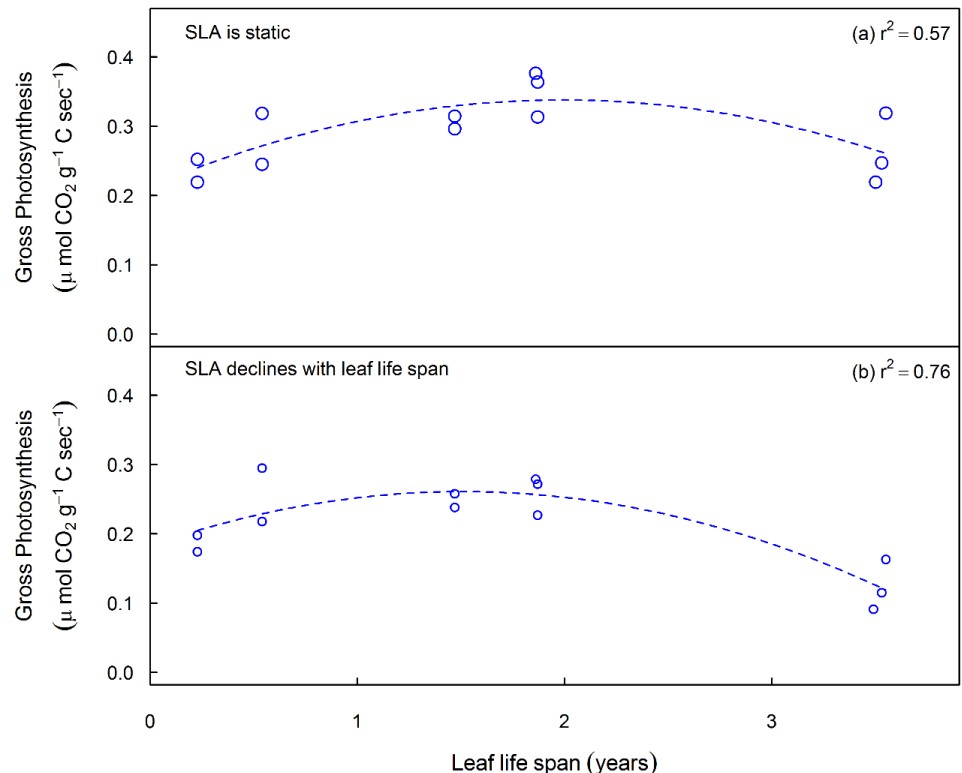









**Figure 8** Relationship of modeled mean annual latex yield and mean annual soil moisture for a rubber plantation over a 10-year period of simulated scenarios. The without drought simulation used the default climate conditions while there were five simulations that considered different types of drought; two simulations assumed drought to occur throughout the year and so these simulations had 20%, 50% lower precipitation than the default precipitation; the other two simulations assumed drought to occur with the extended dry season and so precipitation from April to October was reduced by 30%, 50%, in these simulations; and the final simulation considered shorter dry season but with intense drought so in this simulation precipitation from 8[th] May to 12[th] September was reduced by 50%.

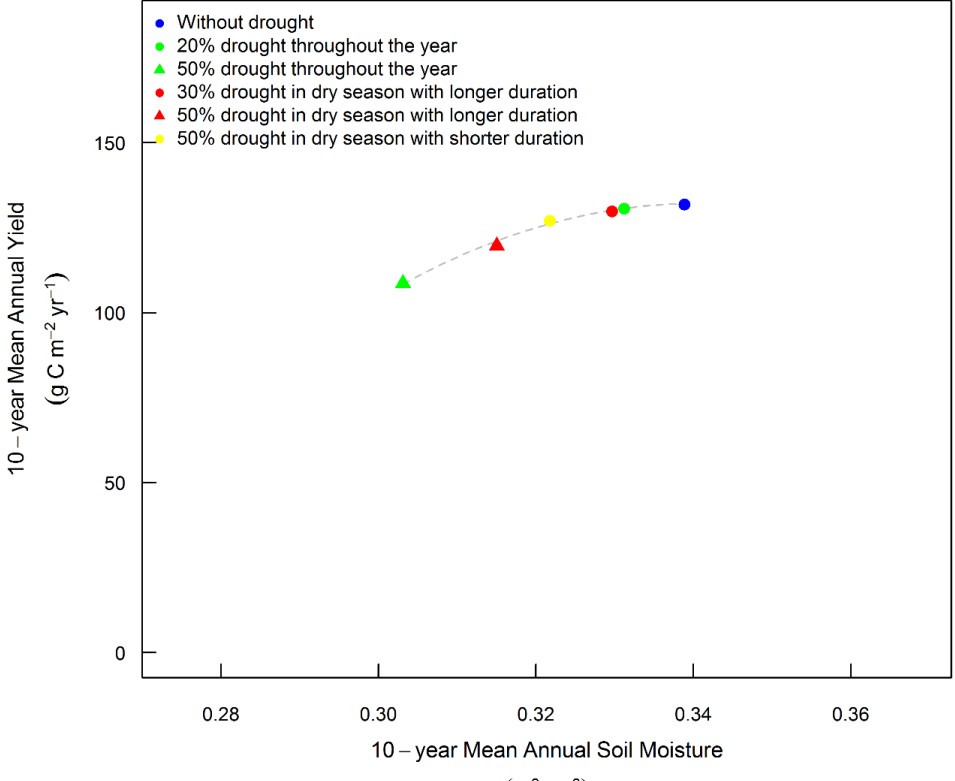





**Tables**
**Table 1** Summary of net ecosystem exchange (NEE = net $CO_2$ uptake), latent (LE) and sensible
(H) heat flux densities, and evapotranspiration (ET) estimates for rubber plantations across
Southeast Asia. The italicized values are estimates derived from the CLM-rubber model.
Negative values indicate a flux toward the land surface (= sink) while positive values indicate a
flux toward the atmosphere (= source). $R_{net}$ is net radiation.

| Location | Mean NEE of the wettest month (kg C m$^{-2}$ yr$^{-1}$) | Mean NEE of the driest month (kg C m$^{-2}$ yr$^{-1}$) | Mean Annual Rainfall (mm yr$^{-1}$) | Mean Annual $R_{net}$ (W m$^{-2}$) | Mean Annual ET (mm yr$^{-1}$) | Mean Annual Latent Heat (W m$^{-2}$) | Mean Annual Sensible Heat (W m$^{-2}$) |
|---|---|---|---|---|---|---|---|
| Xiushuangbanna, China | NA | NA | 1504 | 123.3 | 1125 | 87.4 | NA |
| CRRI, Cambodia | NA | NA | 1439 | 151 | 1459 | 112.5 | NA |
| Som Sanuk, Thailand | -2.35 | 0.68 | 2145 | 129.5 | 1211 | 93.5 | 26.9 |
| This study - Jambi, Indonesia | *-0.25* | *0.09* | 2849 | *139.4* | *964* | *76.4* | *62.9* |

**Table 2** Comparison of water fluxes from CLM-rubber with a soil water model (Kurniawan et
al., 2018) that is parameterized with the site-specific characteristics of the rubber plantations in
the Harapan landscape.

| | CLM-rubber | Soil Water Model |
|---|---|---|
| Transpiration (mm yr$^{-1}$) | 625 | 594 |
| Evapotranspiration (mm yr$^{-1}$) | 964 | 1077 |
| (Runoff + drainage)/Precipitation (unitless) | 0.66 | 0.68 |

**Table 3** Comparison among above ground biomass (AGB) of rubber plantations in the tropics
with similar age.

| | AGB (kg C m$^{-2}$) | Source |
|---|---|---|
| South-West China | 3.92 ± 0.82 | Yang et al. (2016) |
| Western Ghana, Africa | 5.72 ± 0.96 | Wauters et al. (2008) |
| Mato Grosso, Brazil | 3.12 ± 0.72 | Wauters et al. (2008) |
| Harapan Indonesia | 3.36 ± 0.43 | Kotowska et al. (2015) |
| CLM-rubber model for Harapan, Indonesia | 2.98 | This study |
