# Peer review of "Observation-based implementation of ecophysiological processes for a rubber plant"

_Geoscientific Model Development, 2018_

## Referee Comment (RC1) · Anonymous Referee #1 · 6 Dec 2018

Review of "Observation-based implementation of ecophysiological processes for a rubber plant functional type in the community land model (CLM4.5-rubber_v1)"

The authors develop a new plant functional type for the community land model CLM4.5 representing the rubber plant to simulate rubber plantations in the Indonesian province of Jambi. They show that their model can reproduce measurements of plant productivity, biomass, soil carbon, and water fluxes at one site.

The major drawback of this implementation is that the model has been calibrated to

match observations of the study site with specific environmental and management conditions. Initial simulations with a more generic implementation did not produce results that compared well with measurements. Therefore, it remains unclear if their model is suitable to be applied to other regions and management systems. Furthermore, I wonder why the authors initially chose to base their work on CLM which its necessarily (as a land surface scheme for an Earth System Model) crude representations of vegetation structure and processes given their focus on a single site.

More specific comments below.

p. 9, line 214: Please provide a more detailed explanation for your assumption about the effect of tapping on carbon allocation. How did you derive formula 3 from this assumption? Is there any reference literature on the proposed mechanism?

p. 9, line 233: Is there a limit to amount of tapping before a tree wouldn't survive?

p. 11, line 288: Here you say that figure S1 shows values for simulations and measurements of tropical evergreen forest, but the caption of S1 says it shows values for rubber plantations.

p. 11, line 292: What do you mean by "we used the default parametrization of stress deciduous tropical PFT of CLM4.5, but the generic model performance was poor relative to some of the measurements"? Did you apply tapping to the unmodified PFT without the changes in allocation, phenology, and effects of soil water on stomatal conductance described later in the paper?

p. 12, line 333: "pers. comm." with whom?

p. 13, line 335: Please define "brevi-deciduous" as this is no common language (drought-deciduous?).

p. 13, line 361: Please describe in more detail how you determined the revised values of the parameters smpso and smpsc. Here you cite Verhoef and Egea (2014), but this paper does not present a value for rubber or certain tree species. Did you use the

model of Verhoef and Egea in CLM? In this case, how did you determine the required soil hydraulic parameters?

p. 14, line 381: How did you derive the assumption that the base rate of maintenance respiration for fine roots is 50% higher than in the standard model version. Doughty et al. (2018) find higher root respiration at lower fertility sites with lower cation exchange capacities, but don't quantify the size of the effect. However, they don't find a significant difference in maintenance between more and less fertile sites.

p. 14, line 391: Is there any physiological explanation/reference for the implemented additional increase in growth respiration of fine roots? Is this typical for your site or rubber trees in general?

p. 15, line 406: I don't understand your reasoning to increase the Q10 value. Q10 describes the increase of soil respiration per 10°C increase in temperature. You argue that it is necessary to increase Q10, because "rubber plantations at our study sites are 0.5°C hotter than forests". That's not an argument to increase Q10, because higher temperatures will lead to higher soil respiration under the same Q10 value anyway. Do soil characteristics at your test site explain why it should be necessary to double Q10?

p. 16, line 453: "The leaf life span is high in the wet than the dry season." Do you mean higher?

p. 16, line 454: You state that there is no temporal data on rubber tree SLA. How did you derive the values for the month-dependent function?

p. 17, line 466: Delete "the" in "in the Southeast Asia".

p. 17, line 477: You results don't show that CLM-rubber is able to simulate the dynamics of NPP and above-ground biomass, because you compare a simulated time-series with a single measurement. While the simulation agrees with the measurement this does not automatically imply that simulated changes up to this point are also representative of real dynamics. For soil organic carbon you compare two measurements with model

results. While both data sources indicate a decline over time, it is unclear if this change is statistically significant. The same applies to figure S2.

p. 17, line 482: "The modeled biomass of the fine roots and the annual latex yield were much closer to the measurements in the model validation case (Figure S3; a, b) than the model calibration case (Figure 3; a, b)." The big difference between the two figures is not visible.

p. 18, line 495: While Figure 5 shows a good agreement between modelled and measured temporal variability of transpiration, absolute value agree substantially. By around 40% in Fig. 5a and almost 50% in Fig. 5b. This is no close agreement.

p. 18, line 510: Please provide a better explanation of the proposed mechanism and why you base the assumption that "long-lived rubber leaves could have a low mass-based photosynthesis, and that rubber plants could spend carbon in the construction of other tissues such as those associated with protection against insects or prevention of leaf diseases." on model results. It seems you simulated photosynthesis using both fixed and dynamic SLA and then fit a function to simulated monthly average peak photosynthesis. The fit of the function is better (higher r square) for the dynamic SLA case. But how can you validate model results by a function fitted to these results?

p. 19, line 527: Are there no studies available that analyzed the effects of climatic parameters on rubber yields in Indonesia or South East Asia? Something similar to: https://www.sciencedirect.com/science/article/pii/S0168192398000513

p. 19, line 538: Did you really conduct simulations at the ecosystem scale or simulations of the single rubber PFT?

p. 19, line 541: "high" should probably be "higher"

p. 19, line 543: Please explain the data showed in Table 2 in more detail. Are these value for a specific year or average values over a longer timeframe?

p. 21, line 575: Q10 is a soil parameter and has nothing to do with PFTs. Furthermore,

the study of Meyer et al. (2018) is based on measurements in Germany and Belgium so it is questionable if their results are directly transferable to tropical conditions. Meyer et al. also found that Q10 varies with soil moisture, so their main finding is not that models generally use Q10 values that are too low, but using dynamic Q10 values that vary with soil moisture may be a more suitable approach.

p. 22, line 628: Please provide more evidence for your conclusion that can be used CLM-rubber to perform larger-scale simulations within Indonesia given that you calibrated the model to specific site-conditions. How representative are the simulated yield levels for other regions within Indonesia or other site conditions and management regimes? At least you could simulate yields at one other location with Indonesia that is different to your site.

p.23, line 638: Do you plan to integrate rubber plantations also in CLM-FATES?

---

## Referee Comment (RC2) · Anonymous Referee #2 · 17 Dec 2018

Ali et al. make a manual calibration of the drought-deciduous PFT in CLM to represent rubber trees. The development of PFT parameterisations for such tree crops is important, as they are rarely explicitly represented in large-scale models. The basic concept of the work is fairly solid, but I have reservations around the justification for the calibrated values and the general applicability of the results. The major limitation is that some important parts of the parameterisation are highly site-specific, which will substantially limit the potential to easily apply this model to sites outside the low nutrient and climate envelope to which it has been parameterised. Furthermore, many of the

parameterisation choices are only loosely supported and may be implicitly incorporating other factors such as climate biases. I would have preferred to see a parameterisation which incorporated more flexibility for site conditions – particularly if there are any plans to incorporate this more routinely into a trunk version of CLM where users may be unaware of the caveats of this particular PFT. A clear statement in the conclusions and abstract would help avoid this. I suggest that major revisions are required before publication.

Main comments

The model should be run for more sites, either for the calibration or evaluation, or ideally both, in order to give confidence in its applicability, even within the relatively narrow window of low nutrients and phenology for which it has been parameterised. Several sites are mentioned in Table 1. Why not also run the model for these sites, rather than the current loose comparison to the Jambi results?

The paper as it currently stands does not convincingly demonstrate that the new PFT parameterisation performs better than the basic drought deciduous parameterisation for the Jambi site. This would be easily to solve by adding the results for the unparameterized version to all calibration and evaluation figures.

Expanding on the point made in the first paragraph, above, there are three examples of parameterisation which are highly site-specific and limit wider model application: 1) Parameterising leaf longevity to month (line 185) substantially reduces the wider scale applicability of the model. It is fine if it is only intended to apply the model in the location it has been parameterised for, but it is not appropriate if it is intended to apply the model in other locations or for climate change scenarios. Can the longevity not be linked to climate triggers instead? 2) Similarly, increasing root maintenance respiration to account for increased allocation of carbon to alleviate nutrient limitations constrains the model to only be used in these low-nutrient situations. Surely there are studies which have looked at how allocation or respiration varies with nutrient availability that could

Interactive
comment

be used to develop a more general parameterisation? 3) The Q10 tweaking is not very convincing. First off, the justification for the shift in the Q10 is that the rubber plantations used for calibration are $0.5°C$ hotter than forests. In this case the temperature input to the Q10 could logically be raised by $0.5°C$, but this is no justification to change the Q10 itself. Secondly, there are other factors here that could lead to the lower soil carbon. Is litter fall lower for instance, because less productivity goes into growth due to the latex harvest? Or is erosion of soil following clear-cut of the original forest at fault?

I presume that the climate data used was large-scale, rather than a local weather station. In this case, how can the authors be clear that the differences between the observations and the model are down to the parameterisation of the vegetation and not differences between the real site climate and the large-scale climate dataset? There is a real danger that climate biases from a single site are being parameterised into the vegetation response here.

Please make clear whether the soil respiration measurements are total belowground respiration or pure soil respiration (i.e. using root exclusion cores). When reading how they are compared to the model I sometimes think the former and sometimes the latter. This is important for the justification of the parameterisations made.

The section on model projection under drought (lines 522-527) is a bit under-developed. Even setting aside whether the current parameterisation can be expected to perform well under conditions outside the range for which it was parameterised, this section currently doesn't tell us much. It would be better to elaborate mechanistically on how the different scenarios simulated link to different yield reductions – i.e. make some useful hypotheses. Otherwise, why make all these simulations?

Other comments

Line 193. Please explain a bit better the pools here. It is not immediately clear what a "growth pool" is. This might be a pool of labile carbon to be used for growth, or a pool of structural tissue carbon (e.g. wood). I think you mean the latter, but it is not

immediately clear.

Line 228. Why is fcur_st defined as this value? If it is a guess, please say so and explain the logic used.

Line 227. Please provide a citation or explanation for tapping starting at six years.

Line 280. Please explain the principle aspects of the model spin-up and climate data used, so that the reader does not have to refer to another paper to get the basic idea of the simulation set-up.

Line 228. Did you also transfer below-ground carbon to litter following the clear-cut?

Line 333. Pers. comm. from who? Please also be more precise about how this guess was "educated", explaining the logic used.

Line 334. "with a depth of the dip not so large in the dry season", is not clear, please rephrase.

Line 337. The change of LAI in the dry season is not in Fig. 2.

Lines 337-341. Why even explain this experiment? It has not improved the model behaviour and there is no data to support it anyway.

Line 363. Citations or logic needed backing up why these are in the range of plausibility. The comparison to existing ranges in CLM PFTs is not convincing.

Line 490. This logic doesn't hold. There could also be a systematic underestimation of leaf turnover rate.

Lines 554-557. Actually, the effect on the carbon cycle of not getting leaf life span correct is not shown. Please quantify this to support the statement made.

Lines 659-661. This is very tenuously supported by the work here. More confidence in the parameterisation and application of the model at these other sites would be required to support this.

Figure 7. Both panels can be combined into one for easier comparison and to save space.

[Figure]

---

## Author Comment (AC1) · 28 Feb 2019

Review of "Observation-based implementation of ecophysiological processes for a rubber plant functional type in the community land model (CLM4.5-rubber_v1)"

We thank the reviewer for reviewing our work. We acknowledge that our model developments involved several assumptions and thus had some level of uncertainty associated with it in the first instance. Therefore, we have removed the sections that involved

calibrating soil respiration. We place the reviewer's comment as "C" and provide our response as "R".

C1: The authors develop a new plant functional type for the community land model CLM4.5 representing the rubber plant to simulate rubber plantations in the Indonesian province of Jambi. They show that their model can reproduce measurements of plant productivity, biomass, soil carbon, and water fluxes at one site.

R1: We respectfully disagree with the reviewer here. We showed model development and calibration at two different sites, where the differences included age of the plantation, soil texture and management intensity.

C2: The major drawback of this implementation is that the model has been calibrated to match observations of the study site with specific environmental and management conditions. Initial simulations with a more generic implementation did not produce results that compared well with measurements. Therefore, it remains unclear if their model is suitable to be applied to other regions and management systems.

R2: We appreciate this comment very much and thank the reviewer for raising his/her concern. Now, we have performed model evaluations at two contrasting climate zones (Thailand and Cambodia) and with presumably different management practices, high soil fertility. Please see the model evaluation section in the manuscript.

C3: Furthermore, I wonder why the authors initially chose to base their work on CLM which its necessarily (as a land surface scheme for an Earth System Model) crude representations of vegetation structure and processes given their focus on a single site.

R3: We would like to look at how carbon, energy and water fluxes vary spatially over the Jambi Province for different land use systems. We now present our land use map in the manuscript (see Figure S11). The reason why we are representing the processes for rubber in CLM is that in our previous studies, we have developed a sub-canopy

module for oil-palm (Fan et al., 2015) in CLM. To be consistent, we would like to use the same "parent" model and run it on the Jambi province. Indeed, CLM has been used in many land use change studies.

More specific comments below. C4: p. 9, line 214: Please provide a more detailed explanation for your assumption about the effect of tapping on carbon allocation. How did you derive formula 3 from this assumption? Is there any reference literature on the proposed mechanism?

R4: We agree with the reviewer here. We have improved the description of tapping in the manuscript (see lines 231-239) and it is more realistic now. In order to address the effect of tapping on carbon allocation, we have performed the sensitivity analyses key measures of productivities to changes in latex tapping fraction (please see lines 401-407).

C5: p. 9, line 233: Is there a limit to amount of tapping before a tree wouldn't survive?

R5: Yes, in reality if the soil has a low fertility and trees are tapped with either a large amount of intensity or tapped continuously without resting days, then they could eventually die. Currently, we do not consider the "resting days" in the model. Rubber plantations typically have 1 to 2 resting days (Giambelluca et al., 2016), which can be dependent on the age of the plantation and the management.

C6: p. 11, line 288: Here you say that figure S1 shows values for simulations and measurements of tropical evergreen forest, but the caption of S1 says it shows values for rubber plantations.

R6: Thanks for pointing this out. We have corrected it now.

C7: p. 11, line 292: What do you mean by "we used the default parametrization of stress deciduous tropical PFT of CLM4.5, but the generic model performance was poor relative to some of the measurements"? Did you apply tapping to the unmodified PFT without the changes in allocation, phenology, and effects of soil water on stomatal

conductance described later in the paper?

R7: We have removed the text "we used the default parametrization of stress deciduous tropical PFT of CLM4.5, but the generic model performance was poor relative to some of the measurements" from the manuscript now. To facilitate better understanding of the model development and the effect of parameters, including tapping on carbon and water fluxes, we now show comparisons using the parameters of the drought deciduous PFT as it is and the simulations using the rubber parameters.

C8: p. 12, line 333: "pers. comm." with whom? R8: We have removed this text from the manuscript now.

C9: p. 13, line 335: Please define "brevi-deciduous" as this is no common language (drought-deciduous?). R9: We have also removed this text from the manuscript now, but note that previous studies (Giambelluca et al., 2016; e.g. Kotowska et al., 2016) have used this term to describe the phenology of rubber.

C10: p. 13, line 361: Please describe in more detail how you determined the revised values of the parameters smpso and smpsc. Here you cite Verhoef and Egea (2014), but this paper does not present a value for rubber or certain tree species. Did you use the model of Verhoef and Egea in CLM? In this case, how did you determine the required soil hydraulic parameters?

R10: We did not use the model from Verhoef and Egea (2014) and thus have removed this part from the paper. However, we now describe in the manuscript (in lines 380-386) the following: "...by modifying the default soil water potential for drought deciduous tropical PFT in the model for stomatal opening "smpso = -35000 mm" to "smpso = -8750 mm" and stomatal closing "smpsc = -224000 mm" to "smpsc = -56000 mm". To arrive at these values, we first reduced the stomatal slope from 9 to 5 in a step-wise fashion. We could not further reduce the transpiration so we started changing these two parameters. We reduced these two parameters also in a step-wise fashion until the transpiration started to be limiting."

C10: p. 14, line 381: How did you derive the assumption that the base rate of maintenance respiration for fine roots is 50% higher than in the standard model version. Doughty et al. (2018) find higher root respiration at lower fertility sites with lower cation exchange capacities, but don't quantify the size of the effect. However, they don't find a significant difference in maintenance between more and less fertile sites.

R10: We do not focus on calibrating the soil respiration processes so have also removed the text that focused on changing the base rate of maintenance respiration from the manuscript now.

C11: p. 14, line 391: Is there any physiological explanation/reference for the implemented additional increase in growth respiration of fine roots? Is this typical for your site or rubber trees in general?

R11: We decided not to focus anymore on improving the soil respiration and thus have also removed this part from the manuscript now.

C12: p. 15, line 406: I don't understand your reasoning to increase the Q10 value. Q10 describes the increase of soil respiration per 10_C increase in temperature. You argue that it is necessary to increase Q10, because "rubber plantations at our study sites are 0.5_C hotter than forests". That's not an argument to increase Q10, because higher temperatures will lead to higher soil respiration under the same Q10 value anyway. Do soil characteristics at your test site explain why it should be necessary to double Q10?

R12: We do not consider changing Q10 anymore and thus have removed this part from the manuscript now.

C13: p. 16, line 453: "The leaf life span is high in the wet than the dry season." Do you mean higher?

R13: We do not consider seasonal changes in specific leaf area with leaf age and so have removed this section from the manuscript now. Nevertheless, we think that this finding was quite novel, emphasizing the need for a dynamic seasonal component of

functional traits in models.

C14: p. 16, line 454: You state that there is no temporal data on rubber tree SLA. How did you derive the values for the month-dependent function?

R14: We have removed this section from the manuscript now.

C15: p. 17, line 466: Delete "the" in "in the Southeast Asia".

R15: We have eliminated this section from the manuscript.

C16: p. 17, line 477: You results don't show that CLM-rubber is able to simulate the dynamics of NPP and above-ground biomass, because you compare a simulated time-series with a single measurement. While the simulation agrees with the measurement this does not automatically imply that simulated changes up to this point are also representative of real dynamics. For soil organic carbon you compare two measurements with model results. While both data sources indicate a decline over time, it is unclear if this change is statistically significant. The same applies to figure S2.

R16: We agree with the reviewer and now we do not mention that the model was able to 'simulate the dynamics of productivity' but instead say that the model 'captured the measured value'; see lines 585-586. We have removed the soil organic figure from the manuscript. Previous studies (van Straaten et al., 2015) at our studied sites did show that the decline of soil organic carbon was significant.

C17: p. 17, line 482: "The modeled biomass of the fine roots and the annual latex yield were much closer to the measurements in the model validation case (Figure S3; a, b) than the model calibration case (Figure 3; a, b)." The big difference between the two figures is not visible.

R17: We agree with the reviewer. We would like to point out that the soil texture and the age of the plantation differs between the validation and calibration cases.

C18: p. 18, line 495: While Figure 5 shows a good agreement between modelled and

measured temporal variability of transpiration, absolute value agree substantially. By around 40% in Fig. 5a and almost 50% in Fig. 5b. This is no close agreement.

R18: We don't quite understand what the reviewer is saying "absolute value agree substantially". We think now we show better agreements though. This was achieved via re-calibration.

C19: p. 18, line 510: Please provide a better explanation of the proposed mechanism and why you base the assumption that "long-lived rubber leaves could have a low massbased photosynthesis, and that rubber plants could spend carbon in the construction of other tissues such as those associated with protection against insects or prevention of leaf diseases." on model results. It seems you simulated photosynthesis using both fixed and dynamic SLA and then fit a function to simulated monthly average peak photosynthesis. The fit of the function is better (higher r square) for the dynamic SLA case. But how can you validate model results by a function fitted to these results?

R19: We have eliminated this section from the manuscript.

C20: p. 19, line 527: Are there no studies available that analyzed the effects of climatic parameters on rubber yields in Indonesia or South East Asia? Something similar to: https://www.sciencedirect.com/science/article/pii/S0168192398000513

R20: Yes, surely there are but to evaluate the performance of the model, we require other data-sets too, like climate data, for example.

C21: p. 19, line 538: Did you really conduct simulations at the ecosystem scale or simulations of the single rubber PFT?

R21: Yes, we conducted simulations of a single PFT. We did not consider understory vegetation. And, now, we do not compare ecosystem scale fluxes in Jambi.

C22: p. 19, line 541: "high" should probably be "higher"

R22: We have removed this section from the manuscript.

C23: p. 19, line 543: Please explain the data showed in Table 2 in more detail. Are these value for a specific year or average values over a longer timeframe?

R22: We have also removed this section from the manuscript.

C24: p. 21, line 575: Q10 is a soil parameter and has nothing to do with PFTs. Furthermore, the study of Meyer et al. (2018) is based on measurements in Germany and Belgium so it is questionable if their results are directly transferable to tropical conditions. Meyer et al. also found that Q10 varies with soil moisture, so their main finding is not that models generally use Q10 values that are too low, but using dynamic Q10 values that vary with soil moisture may be a more suitable approach.

R24: We do not focus on Q10 anymore so have also removed this section from the manuscript.

C25: p. 22, line 628: Please provide more evidence for your conclusion that can be used CLM-rubber to perform larger-scale simulations within Indonesia given that you calibrated the model to specific site-conditions. How representative are the simulated yield levels for other regions within Indonesia or other site conditions and management regimes? At least you could simulate yields at one other location with Indonesia that is different to your site.

R25: Yes, we have evaluated our model now and put it into our manuscript. Also now we state in the manuscript in lines 811-818 the following "In lowland areas of Jambi, with highly weathered mineral soils, rubber plantations are very rarely fertilized, and soil fertility (i.e. soil N availability, extractable P, and soil organic C) decreased with conversion of forest to rubber (Allen et al., 2015, 2016). Additionally, previous study at the rubber plantations in the Jambi province has shown that it did not differ by a large amount (Kotowska et al., 2015). On the basis that CLM-rubber predicted the rubber yields reasonably well, we think that CLM-rubber can simulate the dynamics of rubber plantations on the Jambi province (see Figure S11)."

C26: p.23, line 638: Do you plan to integrate rubber plantations also in CLM-FATES? R26: Yes, we will certainly put it into CLM5 and CLM5-FATES. We would like to look at the aging processes for rubber in CLM5-FATES.

References Allen, K., Corre, M. D., Tjoa, A. and Veldkamp, E.: Soil Nitrogen-Cycling Responses to Conversion of Lowland Forests to Oil Palm and Rubber Plantations in Sumatra, Indonesia, PLOS ONE, 10(7), e0133325, doi:10.1371/journal.pone.0133325, 2015.

Allen, K., Corre, M. D., Kurniawan, S., Utami, S. R. and Veldkamp, E.: Spatial variability surpasses land-use change effects on soil biochemical properties of converted lowland landscapes in Sumatra, Indonesia, Geoderma, 284, 42–50, doi:10.1016/j.geoderma.2016.08.010, 2016.

Fan, Y., Roupsard, O., Bernoux, M., Le Maire, G., Panferov, O., Kotowska, M. M. and Knohl, A.: A sub-canopy structure for simulating oil palm in the Community Land Model (CLM-Palm): phenology, allocation and yield, Geosci. Model Dev., 8(11), 3785–3800, doi:https://doi.org/10.5194/gmd-8-3785-2015, 2015.

Giambelluca, T. W., Mudd, R. G., Liu, W., Ziegler, A. D., Kobayashi, N., Kumagai, T., Miyazawa, Y., Lim, T. K., Huang, M., Fox, J., Yin, S., Mak, S. V. and Kasemsap, P.: Evapotranspiration of rubber (Hevea brasiliensis) cultivated at two plantation sites in Southeast Asia, Water Resour. Res., 52(2), 660–679, doi:10.1002/2015WR017755, 2016.

Kotowska, M. M., Leuschner, C., Triadiati, T., Meriem, S. and Hertel, D.: Quantifying above- and belowground biomass carbon loss with forest conversion in tropical lowlands of Sumatra (Indonesia), Glob. Change Biol., 21(10), 3620–3634, doi:10.1111/gcb.12979, 2015.

Kotowska, M. M., Leuschner, C., Triadiati, T. and Hertel, D.: Conversion of tropical lowland forest reduces nutrient return through litterfall, and alters nutrient use

efficiency and seasonality of net primary production, Oecologia, 180(2), 601–618, doi:10.1007/s00442-015-3481-5, 2016.

---

## Author Comment (AC2) · 28 Feb 2019

Ali et al. make a manual calibration of the drought-deciduous PFT in CLM to represent rubber trees. The development of PFT parameterisations for such tree crops is important, as they are rarely explicitly represented in large-scale models. The basic concept of the work is fairly solid, but I have reservations around the justification for the calibrated values and the general applicability of the results.

[Figure]

We thank the reviewer for reviewing our work. We acknowledge that our model developments involved several assumptions and thus had some level of uncertainty associated with it in the first instance. Therefore, we have removed the sections that involved calibrating soil respiration. We place the reviewer's comment as "C" and provide our response as "R".

C1: The major limitation is that some important parts of the parameterisation are highly site-specific, which will substantially limit the potential to easily apply this model to sites outside the low nutrient and climate envelope to which it has been parameterised.

R1: We appreciate this comment very much and thank the reviewer for raising his/her concern. Now, we have performed model evaluations at two contrasting climate zones (Thailand and Cambodia sites) and with presumably different management practices, including high soil fertility. Please see the model evaluation section in the manuscript.

C2: Furthermore, many of the parameterisation choices are only loosely supported and may be implicitly incorporating other factors such as climate biases. I would have preferred to see a parameterization which incorporated more flexibility for site conditions – particularly if there are any plans to incorporate this more routinely into a trunk version of CLM where users may be unaware of the caveats of this particular PFT. A clear statement in the conclusions and abstract would help avoid this. I suggest that major revisions are required before publication.

R2: We agree with the reviewer. Data limitation is one challenge that we face for the model development, and thus we acknowledge that there is a considerable level of uncertainty in the present model development. We have made a statement in the conclusion as well as in the abstract, where we state now that we "develop and calibrate a site-specific rubber plant functional type (PFT) for the conditions in Indonesia under low soil fertility". Yes, we would like to put this model into the trunk version of CLM but we will do additional tests first.

Main comments C3: The model should be run for more sites, either for the calibration

Interactive
comment

or evaluation, or ideally both, in order to give confidence in its applicability, even within the relatively narrow window of low nutrients and phenology for which it has been parameterised. Several sites are mentioned in Table 1. Why not also run the model for these sites, rather than the current loose comparison to the Jambi results?

R3: Yes, now we have performed model evaluations at two contrasting climate zones (Cambodia and Thailand) and with different management practices and high soil fertility. Please see the model evaluation section in the manuscript.

C4: The paper as it currently stands does not convincingly demonstrate that the new PFT parameterisation performs better than the basic drought deciduous parameterisation for the Jambi site. This would be easily to solve by adding the results for the unparameterized version to all calibration and evaluation figures.

R4: We have now compared the model simulations using the parameters of the default drought deciduous PFT with the simulations of the rubber PFT in a step-by-step format. These results have been incorporated into the manuscript now. The new rubber PFT performs better for predicting leaf area index, net primary productivity, transpiration, leaf-litterfall, root biomass.

Expanding on the point made in the first paragraph, above, there are three examples of parameterisation which are highly site-specific and limit wider model application:

C5: Parameterising leaf longevity to month (line 185) substantially reduces the wider scale applicability of the model. It is fine if it is only intended to apply the model in the location it has been parameterised for, but it is not appropriate if it is intended to apply the model in other locations or for climate change scenarios. Can the longevity not be linked to climate triggers instead?

R5: We tried considering leaf longevity as a function of climate but we could not get the seasonal dynamics as measured. Thus, we now mention in the manuscript the following: "We acknowledge that currently the seasonal variation in leaf longevity is

not a function of climate (e.g. soil water potential, minimum rainfall) and has been developed for a location in Jambi, Indonesia. So we currently suggest adjusting this function to the seasonality of the dry/wet season at other locations."

C6: Similarly, increasing root maintenance respiration to account for increased allocation of carbon to alleviate nutrient limitations constrains the model to only be used in these low-nutrient situations. Surely there are studies which have looked at how allocation or respiration varies with nutrient availability that could be used to develop a more general parameterisation?

R6: We no longer focus on increasing root maintenance respiration and thus have removed this section from the manuscript.

C7: The Q10 tweaking is not very convincing. First off, the justification for the shift in the Q10 is that the rubber plantations used for calibration are 0.5_C hotter than forests. In this case the temperature input to the Q10 could logically be raised by 0.5_C, but this is no justification to change the Q10 itself. Secondly, there are other factors here that could lead to the lower soil carbon. Is litter fall lower for instance, because less productivity goes into growth due to the latex harvest? Or is erosion of soil following clear-cut of the original forest at fault?

R7: We agree that tweaking Q10 is not very convincing and therefore we have removed this section from the manuscript too.

C8: I presume that the climate data used was large-scale, rather than a local weather station. In this case, how can the authors be clear that the differences between the observations and the model are down to the parameterisation of the vegetation and not differences between the real site climate and the large-scale climate dataset? There is a real danger that climate biases from a single site are being parameterised into the vegetation response here.

R8: We have made it clear in the manuscript that local weather climate data was used

from the clear-cut condition at all studied sites.

C9: Please make clear whether the soil respiration measurements are total below-ground respiration or pure soil respiration (i.e. using root exclusion cores). When reading how they are compared to the model I sometimes think the former and sometimes the latter. This is important for the justification of the parameterisations made.

R9: Soil respiration is the total belowground respiration, which includes the autotrophic and heterotrophic respiration.

C10: The section on model projection under drought (lines 522-527) is a bit underdeveloped. Even setting aside whether the current parameterisation can be expected to perform well under conditions outside the range for which it was parameterised, this section currently doesn't tell us much. It would be better to elaborate mechanistically on how the different scenarios simulated link to different yield reductions – i.e. make some useful hypotheses. Otherwise, why make all these simulations?

R10: We have removed this section completely from the manuscript.

Other comments C11: Line 193. Please explain a bit better the pools here. It is not immediately clear what a "growth pool" is. This might be a pool of labile carbon to be used for growth, or a pool of structural tissue carbon (e.g. wood). I think you mean the latter, but it is not immediately clear.

R11: We have removed this statement from the manuscript.

C12: Line 228. Why is fcur_st defined as this value? If it is a guess, please say so and explain the logic used.

R11: We have also removed the use of "fcur_st". We have improved the description of tapping in the manuscript (see lines 231-239) and it is more realistic now.

C13: Line 227. Please provide a citation or explanation for tapping starting at six years.

R13: We have provided the citation now. See lines 284-286 in the manuscript.

[Figure]

C14: Line 280. Please explain the principle aspects of the model spin-up and climate data used, so that the reader does not have to refer to another paper to get the basic idea of the simulation set-up. R14: We have explained the procedure for spinups in the following lines 291-298; specifically, we make a comment as follows: "We performed a model spin-up in two stages; first a pre-industrial simulation spin-up was carried out, where we used 1900's climate data (1900 to 1940) with pre-industrial $CO_2$ concentration for Jambi from CRUNCEP data (Lawrence et al., 2007). We cycled these 40 years of climate data and ran the model for 1000 years in the accelerated mode. Then we ran the model for 500 years in the normal mode to get to the equilibrium state of ecosystem. Next, we carried out the transient simulation from 1941 to 1997, where the climate data is again extracted from CRUNCEP data and historical $CO_2$ concentrations is used".

C15: Line 228. Did you also transfer below-ground carbon to litter following the clearcut?

R15: Yes, we transferred the fineroot carbon and nitrogen into fast litter pools. We assumed that farmers remove the coarse roots from the forest so that they could plant and thus we did not move them into any litter pools. We have stated these in lines 305-307.

C16: Line 333. Pers. comm. from who? Please also be more precise about how this guess was "educated", explaining the logic used.

R16: We have also removed this statement.

C17: Line 334. "with a depth of the dip not so large in the dry season", is not clear, please rephrase.

R17: We have removed this statement.

C18: Line 337. The change of LAI in the dry season is not in Fig. 2.

R18: We have removed this statement.

C19: Lines 337-341. Why even explain this experiment? It has not improved the model behaviour and there is no data to support it anyway.

R19: We would like to keep this experiment in the manuscript because it shows that we considered changing the critical value of soil water potential.

C20: Line 363. Citations or logic needed backing up why these are in the range of plausibility. The comparison to existing ranges in CLM PFTs is not convincing.

R20: We have reshaped the focus of the manuscript and pointed out that the assumptions are likely to have uncertainty associated with it.

C21: Line 490. This logic doesn't hold. There could also be a systematic underestimation of leaf turnover rate.

R21: We disagree with the reviewer here and argue that it can be possible if not all of the rubber trees drop their leaves at the same-time in the dry season.

C22: Lines 554-557. Actually, the effect on the carbon cycle of not getting leaf life span correct is not shown. Please quantify this to support the statement made.

R22: We have removed this statement.

C23: Lines 659-661. This is very tenuously supported by the work here. More confidence in the parameterisation and application of the model at these other sites would be required to support this.

R23: We have also removed this statement.

C24: Figure 7. Both panels can be combined into one for easier comparison and to save space. R24: We have removed this figure too. But we still think that the results of this figure was quite novel, emphasizing the need for a dynamic seasonal component of functional traits in models.

References

Lawrence, D. M., Thornton, P. E., Oleson, K. W. and Bonan, G. B.: The Partitioning of Evapotranspiration into Transpiration, Soil Evaporation, and Canopy Evaporation in a GCM: Impacts on Land–Atmosphere Interaction, J. Hydrometeorol., 8(4), 862–880, doi:10.1175/JHM596.1, 2007.